# Calibration and evaluation of CCD spectroradiometers for ground-based and airborne measurements of spectral actinic flux densities

Birger Bohn[1] and Insa Lohse[1,2]

[1]Institut für Energie- und Klimaforschung, IEK-8: Troposphäre, Forschungszentrum Jülich GmbH, 52428 Jülich, Germany
[2]Deutscher Wetterdienst, BTZ-Langen, 63225 Langen, Germany

*Correspondence to:* B. Bohn (b.bohn@fz-juelich.de)

**Abstract.** The properties and performance of CCD array spectroradiometers for the measurement of atmospheric spectral actinic flux densities (280–650 nm) and photolysis frequencies were investigated. These instruments are widely used in atmospheric research and are suitable for aircraft applications because of high time resolutions and high sensitivities in the UV range. The laboratory characterization included instrument-specific properties like wavelength accuracy, dark signals, dark noise and signal-to-noise ratios. Spectral sensitivities were derived from measurements with spectral irradiance standards. The calibration procedure is described in detail and a straightforward method to minimize the influence of stray light on spectral sensitivities is introduced. From instrument dark noise, minimum detection limits $\approx 1 \times 10^{10} \mathrm{cm}^{-2} \mathrm{s}^{-1} \mathrm{nm}^{-1}$ were derived for spectral actinic flux densities at wavelengths around 300 nm (1 s integration time). As a prerequisite for the determination of stray light under field conditions, atmospheric cutoff wavelengths were defined using radiative transfer calculations as a function of solar zenith angles and ozone columns. The recommended analysis of field data relies on these cutoff wavelengths and is also described in detail taking data from a research flight on HALO as an example. An evaluation of field data was performed by ground-based comparisons with a double-monochromator based, highly sensitive reference spectroradiometer. Spectral actinic flux densities were compared as well as photolysis frequencies $j(\mathrm{NO_2})$ and $j(\mathrm{O^1D})$, representing UV-A and UV-B ranges, respectively. The spectra expectedly revealed increased daytime levels of stray light induced signals and noise below atmospheric cutoff wavelengths. The influence of instrument noise and stray light induced noise was found to be insignificant for $j(\mathrm{NO_2})$ and rather limited for $j(\mathrm{O^1D})$, resulting in estimated detection limits of $5 \times 10^{-7} \mathrm{s}^{-1}$ and $1 \times 10^{-7} \mathrm{s}^{-1}$, respectively, derived from nighttime measurements on the ground (0.3 s integration time, 10 s averages). For $j(\mathrm{O^1D})$ the detection limit could be further reduced by setting spectral actinic flux densities to zero below atmospheric cutoff wavelengths. The accuracies of photolysis frequencies were determined from linear regressions with data from the double-monochromator reference instrument. The agreement was typically within $\pm 5\%$. Because optical receiver aspects are not specific for the CCD spectroradiometers they were widely excluded in this work and will be treated in a separate paper in particular with regard to airborne applications.

## 1 Introduction

Solar actinic radiation is the driving force of atmospheric photochemistry because it produces short-lived reactive radicals in photolysis processes. Photolysis frequencies are first-order rate constants that quantify the corresponding loss rate of photolizable compounds in the gas phase. Likewise they determine the primary production rate of the often highly reactive photolysis products like $O(^1D)$, $O(^3P)$ or OH radicals. Photolysis frequencies are therefore essential parameters for a quantitative understanding of atmospheric photochemistry.

Taking formation of $O(^1D)$ in the photolysis of ozone as an example,

$$O_3 + h\nu(\lambda \leq 340 \text{ nm}) \longrightarrow O_2 + O(^1D) \tag{R1}$$

the corresponding photolysis frequency $j(O^1D)$ is defined by:

$$\frac{d[O(^1D)]}{dt} = j(O^1D)[O_3] \tag{1}$$

Species in square brackets in Eq. 1 denote gas-phase number concentrations. The photolysis frequency $j(O^1D)$ is of particular importance because in the presence of water vapour the electronically excited $O(^1D)$ can form OH radicals, the primary atmospheric oxidant:

$$O(^1D) + H_2O \longrightarrow 2\text{ OH} \tag{R2}$$

The connection between the chemical rate constant $j(O^1D)$ and the local solar radiation field is given via the spectral actinic flux density $F_\lambda$:

$$j(O^1D) = \int F_\lambda \times \sigma_{O_3} \times \phi_{O(^1D)} \, d\lambda \tag{2}$$

In this equation $\sigma_{O_3}$ and $\phi_{O(^1D)}$ are molecular parameters of $O_3$, namely the absorption cross sections of the precursor molecule and the quantum yields of the photo-product $O(^1D)$, respectively, which confine the process mainly in the UV-B range. $F_\lambda$ is inserted in corresponding molecular units ($\text{cm}^{-2}\text{s}^{-1}\text{nm}^{-1}$).

Other photolysis frequencies can be determined similarly by inserting the respective molecular parameters. For example, in the case of the mainly UV-A driven $NO_2$ photolysis:

$$NO_2 + h\nu(\lambda \leq 420 \text{ nm}) \longrightarrow NO + O(^3P) \tag{R3}$$

the photolysis frequency $j(NO_2)$ is calculated by:

$$j(NO_2) = \int F_\lambda \times \sigma_{NO_2} \times \phi_{O(^3P)} \, d\lambda \tag{3}$$

Accordingly, the most versatile method to determine photolysis frequencies is spectroradiometry: the spectral actinic flux density $F_\lambda$ is measured in the relevant UV/VIS spectral range and any photolysis frequency can be calculated, provided the corresponding molecular parameters $\sigma$ and $\phi$ are known (Hofzumahaus et al., 1999). Other methods like chemical actinometry

and filter radiometry have the disadvantage that they are process specific, but they can be advantageous for other reasons, e.g. for their absolute accuracy and easiness of maintenance for long-term operation, respectively (Kraus et al., 2000; Shetter et al., 2003; Hofzumahaus et al., 2004; Hofzumahaus, 2006). A further advantage of spectroradiometry is that the temperature and pressure dependencies of photolysis frequencies are obtained directly by taking into account the respective dependencies of the molecular parameters. This is particularly important for aircraft measurements where ambient conditions are most variable.

The major technical difficulties related with the radiometric determination of $F_\lambda$ in the atmosphere are (i) the quality of optical receivers for actinic radiation (with an ideally $4\pi$ and angle-independent reception characteristics) and (ii) the accuracy of measurements in the UV-B range that can be affected by low detector sensitivities and non-regularly reflected radiation within monochromators (stray light). Both aspects are particularly challenging for aircraft measurements:

(i) In contrast to ground-based operations where measurements of upward radiation in the UV range may be dispensable under conditions of low ground albedos, aircraft deployments require separate measurements in the upper and the lower hemisphere. Because the $2\pi$ optical receivers for actinic radiation have a vertical extension and limited horizontal shielding, cross-talks to the opposite hemispheres are unavoidable. These cross-talks and imperfections of the receivers in general require specific corrections. Since these corrections are complex and independent of the type of spectroradiometer, we attend to this difficulty in an accompanying paper where wavelength dependent correction factors are derived as a function of time, altitude and atmospheric conditions (Lohse and Bohn, 2017).

(ii) UV-B radiation in the troposphere and the lower stratosphere is strongly diminished by stratospheric ozone. Nevertheless, the remainder is extremely important for atmospheric chemistry because it can photolyze tropospheric ozone and form $O(^1D)$ (R1). To quantify the corresponding photolysis frequency $j(O^1D)$, accurate measurements of spectral actinic flux densities in the UV-B range are required (Eq. 2). Double-monochromator based spectroradiometers have excellent stray light suppression and high sensitivity for measurements of $j(O^1D)$ (Hofzumahaus et al., 1999; Shetter and Müller, 1999). However, the instruments have low time resolutions on the order of 0.5–2 min because the two monochromators have to be scanned synchronously to obtain a spectrum. This is a major drawback for high speed aircraft measurements where conditions can change rapidly through the influence of clouds, changing ground albedo or flight maneuvers. A time resolution on the order of a second is therefore desired. Such a time resolution is achieved by single-monochromator based CCD detector array spectroradiometers (CCD-SR) (Eckstein et al., 2003; Jäkel et al., 2007; Petropavlovskikh et al., 2007; Stark et al., 2007). These instruments have the further advantage of a small size and weight, and of higher mechanical stability because they usually contain no motorized parts. On the other hand, measurements in the UV-B range suffer from an increased level of stray light that is typical for single-monochromator applications (Hofzumahaus, 2006). Therefore, a thorough treatment of stray light induced effects is a prerequisite for high quality measurements in the UV-B range with high time resolution.

In this work we describe the properties, the calibration and the data analysis of CCD-SR based instruments for airborne measurements of spectral actinic flux densities and photolysis frequencies. The equipment was already employed on the research aircraft HALO (High Altitude and Long Range Research Aircraft) and on the Zeppelin NT during several missions. The applied type of CCD-SR is widely used for the determination of atmospheric photolysis frequencies. The instruments can be purchased ready for use and can directly provide photolysis frequencies. However, applications like airborne measurements

require a more complicated post-flight data analysis. Moreover, quality assurance in any case requires regular re-calibrations based on as possible simple guidelines. We will therefore explain in detail how our instruments were calibrated and introduce a straightforward, reliable method how to deal with stray light influence in both laboratory calibrations and in the analysis of field data. In particular, for laboratory calibrations we adopt a new correction factor for subtracted stray light signals obtained with cutoff filters and an optimization of spectral sensitivities using extended integration times. For the evaluation of field data we introduce atmospheric cutoff wavelengths from radiative transfer calculations to define safe, condition dependent wavelength ranges for stray light determination. In addition the precision of $j(\mathrm{O}^1\mathrm{D})$ measurements was improved by excluding spectral actinic flux densities below the cutoff wavelengths. These procedures are thought as recommendations for other users of similar instruments in order to raise the awareness for important instrument properties and characterizations, to illustrate essential evaluation steps and to clarify current limitations. Careful attention is thought to improve data quality and reproducibility. To evaluate the approach, example data from a flight on HALO, as well as ground based comparisons with a double-monochromator reference instrument will be shown.

## 2 Instrument properties

### 2.1 Spectroradiometers and data acquisition

The CCD spectroradiometers and optical receivers used in this work were developed by Meteorologie Consult GmbH (Metcon) specifically for measurements of spectral actinic flux densities and photolysis frequencies in the atmosphere. Five different instruments that are identified by five-digit serial numbers were characterized and calibrated in the laboratory and pairwise used during several deployments on a Zeppelin-NT and on HALO between 2007 and 2015, as well as separately for occasional ground-based measurements. All instruments are similar in construction but exhibit somewhat variable individual characteristics. A slightly modified spectroradiometer of the same type was described previously by Jäkel et al. (2007).

The spectroradiometers are composed of combinations of a monochromator (Zeiss, MCS) and a back-thinned CCD detector array (Hamamatsu, S7031-0906S, windowless), as well as operating electronics (tec5 AG) that are built into air-tight aluminium housings (approx. 30 cm ×15 cm×10 cm). These housings are connected with separate CCD Peltier cooling electronics and power supplies. Radiation is passed into the monochromators by 600 $\mu$m optical fiber feed-throughs terminated with SMA adapters. An actinic $2\pi$ receiver optic can be attached tightly to the instrument housings or more flexibly via optical fibers with user-defined length. The latter option was chosen in this work because it is more convenient for aircraft installations. Optical fiber lengths ranged between 2 m and 12 m. Dependent on the application, one or two spectroradiometers including electronics and a compact computer were built into 19 inch rack-mounts for operation in instrument flight racks. The basic components of a single instrument are shown in Fig. 1. More information on the optical receiver properties and related aircraft specific aspects will be described in a separate paper (Lohse and Bohn, 2017).

Computer communication is established via USB or ethernet interfaces (tec5 AG). CCD data acquisition is controllable by purpose-built software provided by Metcon. The recorded signals ($S$) are dimensionless 16-bit signal counts or so-called analog-to-digital units (ADU) ranging between 0 and 65535. One of the features implemented in the software is the option to

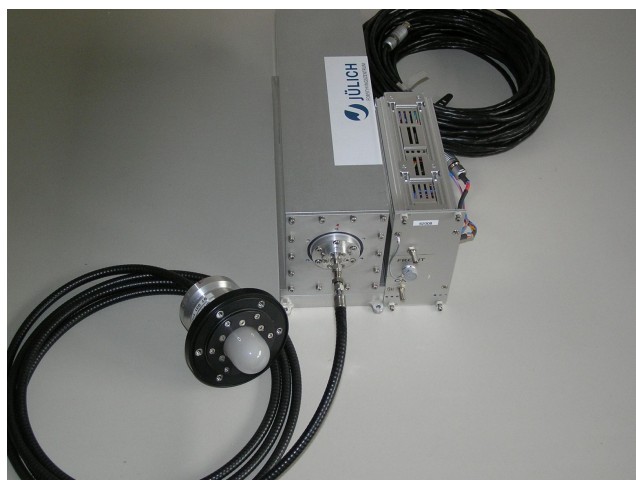

**Figure 1.** Basic components of a CCD array spectroradiometer system for airborne measurements. $2\pi$ receiver optics with optical fiber (black, front), spectrometer including monochromator and CCD array detector (middle), and cooling electronics (right).

measure with different integration times quasi-simultaneously. Typically 4–5 integration times between 3 ms and 300 ms were used in field measurements and up to 1000 ms during laboratory calibrations. Longer integration times are advantageous at low radiation levels to increase the signal-to-noise ratios (Sec. 2.2.2). On the other hand, shorter integration times may be necessary in parts of the spectra to avoid saturation of the CCD detectors. The idea is to combine spectra with different integration times

5 to an optimized spectrum with as long as possible integration times in different wavelength ranges. This optimization is useful as long as integration times are short compared to the time scale of changes of measured flux densities. Moreover, the linearity of the CCD detector is a further requirement that can be tested in the laboratory (Sect. 2.2.3). The spectrum optimization as well as the calculation of actinic flux density spectra and photolysis frequencies is operable already during the measurements by the Metcon software. However, although this prompt analysis is useful for some applications, a post-flight data analysis

is required for airborne measurements by taking into account aircraft locations, rotation angles of the aircraft, and ambient conditions as explained in more detail in Sect. 3.1.

## 2.2 Laboratory characterization and instrument calibration

### 2.2.1 Spectral range, resolution and wavelength accuracy

The employed monochromator type has a ceramic housing with a very low temperature drift ($<0.01$ nm/K). The thermostated

CCD detector is directly fixed to this housing making the setup mechanically insensitive to external temperature drifts. The holographic grating is blazed for 250 nm for optimum UV detection. Radiation enters the monochromator through a cross section converter and is dispersed onto the $532 \times 64$ pixel CCD array. By binning-operation the two-dimensional array is effectively used as a linear sensor array. The wavelength range covered by the spectroradiometers is roughly 260–660 nm with

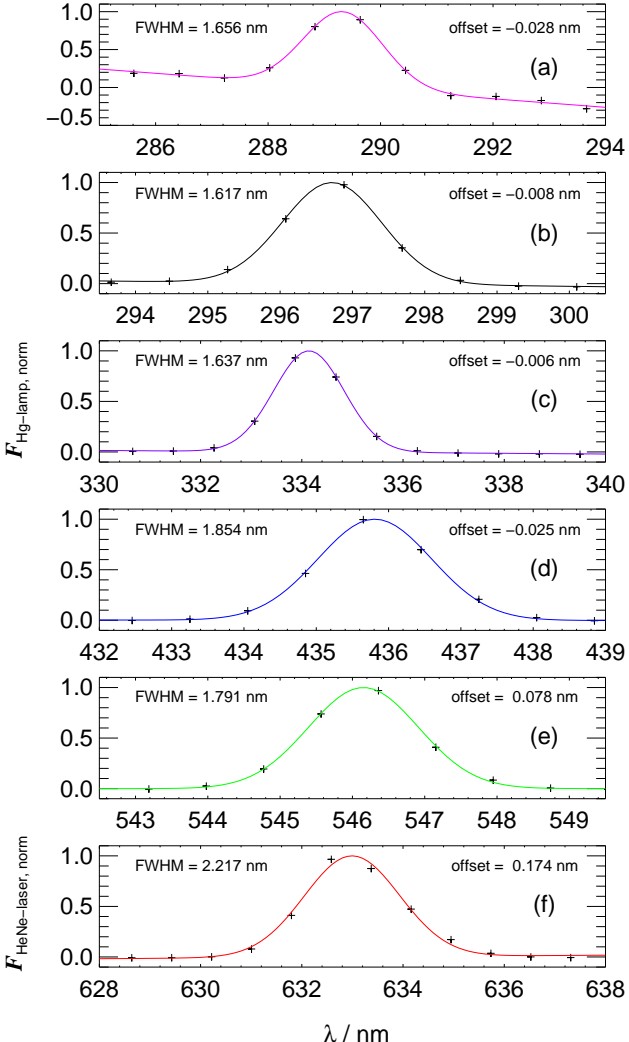

**Figure 2.** Selected sections of background corrected, normalized low-pressure Hg-lamp spectra (a)–(e) and of a HeNe laser spectrum (f), obtained with instrument 62001 and corresponding emission line fits of Eq. 4 (full lines). Wavelengths correspond to manufacturer based third order polynomials of pixel numbers. Indicated wavelength offsets and FWHMs of emission lines were obtained from parameters $a_1$ to $a_3$ of Eq. 4. Emission lines in figures (a)–(f) correspond to in-air wavelengths listed in the first column of Tab. 1.

**Table 1.** Wavelength offsets and slit function half-widths (FWHM) of spectroradiometers obtained by fitting Eq. 4 to Hg emission lines of a low pressure mercury lamp. Dependent on instrument, uncertainties correspond to standard deviations from 3–13 measurements during 3–9 year periods. Data at 632.8 nm were obtained with a HeNe laser instead of a mercury lamp (2–3 measurements during a three year period). Examples of emission line fits are shown in Fig. 2.

| emission- | instrument | | | | |
|---|---|---|---|---|---|
| line / nm | 45853 | 62000 | 62001 | 62008 | 85235 |
| | offset / nm | | | | |
| 289.360 | $-0.241\pm0.041$ | $0.154\pm0.019$ | $0.009\pm0.028$ | $0.032\pm0.016$ | $0.008\pm0.025$ |
| 296.728 | $-0.172\pm0.010$ | $0.170\pm0.009$ | $0.018\pm0.021$ | $0.051\pm0.016$ | $0.014\pm0.024$ |
| 334.148 | $-0.042\pm0.026$ | $0.258\pm0.010$ | $0.021\pm0.023$ | $0.096\pm0.022$ | $0.027\pm0.023$ |
| 435.834 | $0.051\pm0.006$ | $0.475\pm0.007$ | $0.004\pm0.024$ | $0.042\pm0.024$ | $0.053\pm0.022$ |
| 546.075 | $0.071\pm0.005$ | $0.361\pm0.010$ | $0.105\pm0.021$ | $0.086\pm0.025$ | $0.168\pm0.022$ |
| 632.816[a] | $0.050\pm0.018$ | $-0.393\pm0.014$ | $0.162\pm0.010$ | $0.228\pm0.054$ | $0.297\pm0.075$ |
| | FWHM / nm | | | | |
| 289.360 | $2.03\pm0.23$ | $1.82\pm0.10$ | $1.68\pm0.05$ | $1.57\pm0.06$ | $1.64\pm0.03$ |
| 296.728 | $1.86\pm0.04$ | $1.93\pm0.06$ | $1.62\pm0.02$ | $1.58\pm0.01$ | $1.59\pm0.01$ |
| 334.148 | $1.54\pm0.03$ | $2.31\pm0.08$ | $1.65\pm0.02$ | $1.68\pm0.02$ | $1.62\pm0.02$ |
| 435.834 | $1.68\pm0.04$ | $2.84\pm0.10$ | $1.86\pm0.03$ | $1.98\pm0.03$ | $1.67\pm0.01$ |
| 546.075 | $1.71\pm0.04$ | $2.41\pm0.08$ | $1.80\pm0.01$ | $1.76\pm0.01$ | $1.82\pm0.01$ |
| 632.816[a] | $1.57\pm0.01$ | $1.74\pm0.01$ | $2.14\pm0.07$ | $2.02\pm0.10$ | $2.95\pm0.02$ |

[a] HeNe laser measurements

a mean spectral pixel distance of 0.8 nm. The relationship between the CCD pixel number (0–531) and the wavelength is determined by manufacturer-based, instrument-specific third-order polynomial functions.

In order to verify the wavelength positions, spectra of a low pressure mercury pencil-lamp (Oriel) were recorded. After averaging over 50 single measurements, subtracting separately measured dark spectra obtained upon covering optical receivers (Sec. 2.2.2), and application of spectral sensitivities (Sec. 2.2.3), selected Hg emission lines in a range 290–550 nm were fitted with an empirical function $A(\lambda)$ to obtain instrument response functions and wavelength offsets:

$$A(\lambda) = a_0 \, \exp(-a_2(\lambda - a_1)^{a_3}) + B(\lambda) \tag{4}$$

Examples of emission line fits are shown in Fig. 2. The linear function $B(\lambda)$ allows to adjust for a tilted lamp background (lamp-specific continuous emission) and the parameter $a_0$ defines the fitted maximum of the line. The parameters $a_2$ and $a_3$ can be adjusted to match variable line shapes while the parameter $a_1$ denotes the central wavelength of a line with regard to the manufacturer-based wavelengths. The differences between $a_1$ and in-air line positions from the literature (Sansonetti et al., 1996) are defined as wavelength offsets. Moreover, from the parameters $a_2$ and $a_3$ the full width at half maximum (FWHM) can

be determined. Table 1 shows a summary of wavelength offsets and FWHM obtained with the five instruments, that typically do not exceed 0.2 nm and 2 nm, respectively. In addition to the mercury lamp, a HeNe laser (Spectra Physics) was occasionally used as reference line emitter at 632.8 nm.

The quality of the results in Tab. 1 is limited by the relatively small number of data points that represent an emission line. Moreover, the assumption of symmetrical response functions and a linearly changing background may not strictly apply. Nevertheless, the reproducibility of the results is high, indicating that both parameters can be determined within $\pm 0.05$ nm. Wavelength shifts induced by reduced cabin pressures ($\geq$750 mbar) during airborne measurements are considered insignificant ($\leq$0.02 nm at 300 nm).

### 2.2.2 Dark signals, noise and signal-to-noise ratios

When optical receivers are covered and no radiation enters the spectroradiometers, dark signals $S_{\mathrm{dark}}$ can be recorded. As was already described by Jäkel et al. (2007), the dark signal for each pixel is composed of an electronic offset $S_{\mathrm{dark,0}}$ and a thermally induced dark current signal that increases with temperature and integration time. While the electronic offsets are fairly constant, dark currents are slightly different for each pixel but, except for noise, reproducible under temperature-controlled conditions. By averaging over, e.g. 100 single measurements, noise can be reduced and mean dark spectra for each integration time are obtained. Subtracting these dark spectra effectively removes the dark current-induced spectral structure that is underlying all measured spectra. This is particularly important under low-signal conditions. Examples of dark spectra for different integration times are shown in Fig. 3.

Because a slow, permanent change of dark signals with time cannot be excluded, it is useful to update dark spectra regularly. In addition dark signals may be subject to fluctuations that can be caused by external temperature changes or instabilities of the temperature control. The difference between a single measurement of the dark signal and the averaged dark signal may therefore deviate from zero more strongly than expected from the noise of the measurements. For atmospheric measurements this poses no problem because the remaining positive or negative background is determined together with a stray light induced background for each spectrum separately. This will be explained in more detail in Sect. 3.1. The main purpose to subtract mean dark spectra therefore is to obtain an approximate dark correction and to widely remove the dark current induced spectral structures visible in Fig. 3. This feature is also implemented in the Metcon software.

The noise of the dark signals $N_{\mathrm{dark}}$ was determined for each pixel by deriving standard deviations of repeated dark measurements. Table 2 lists the mean dark noise as a function of integration time for the employed instruments. The given ranges correspond to the variations of the noise in the wavelength range 280–650 nm. These ranges are small which means that all CCD pixels exhibit similar noise levels. The increase of noise with dark current signals or integration times on average follows a square root dependence as expected for thermally induced shot noise ($N_{\mathrm{d}}$). This is shown in panel (a) of Fig. 4 for instrument 62001 as an example. The remaining noise towards zero integration time ($N_{\mathrm{r}}$) is considered as a combination of instrument

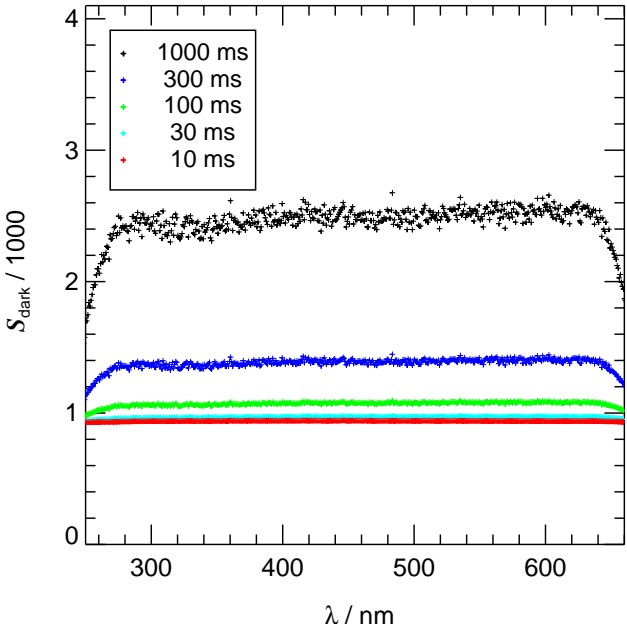

**Figure 3.** Mean dark signal spectra $S_{\mathrm{dark}}$ of instrument 62001 at different integration times. Dark signals are composed of an electronic background ($S_{\mathrm{dark},0} \approx 900$ for this instrument) and a dark current signal that increases linearly with integration time. Pixel-to-pixel variations represent reproducible structures. The corresponding mean noise of the dark signals is comparatively small and listed in Tab. 2

**Table 2.** Mean noise $N_{\mathrm{dark}}$ of spectroradiometer dark signals as a function of integration time and corresponding standard deviations of $N_{\mathrm{dark}}$ in a wavelength range 280–650 nm. Examples of dark spectra are shown in Fig. 3.

|  | integration time / ms | | |
| --- | --- | --- | --- |
| instrument | 10 | 100 | 1000 |
| 45853 | 7.2±0.9 | 7.3±1.0 | 8.3±1.1 |
| 62000 | 6.0±0.6 | 6.5±0.7 | 10.4±1.1 |
| 62001 | 7.1±0.7 | 7.2±0.7 | 11.4±1.3 |
| 62008 | 3.5±0.4 | 4.3±0.5 | 8.7±2.6 |
| 85235 | 2.0±0.2 | 2.3±0.3 | 4.4±1.6 |

specific read-out noise and other off-chip noise, i.e. for the dark noise the following relations apply:

$$
\begin{aligned}
N_{\mathrm{dark}}^2 &\approx N_{\mathrm{d}}^2 + N_{\mathrm{r}}^2 \\
N_{\mathrm{d}} &\propto \sqrt{(S_{\mathrm{dark}} - S_{\mathrm{dark},0})}
\end{aligned}
\tag{5}
$$

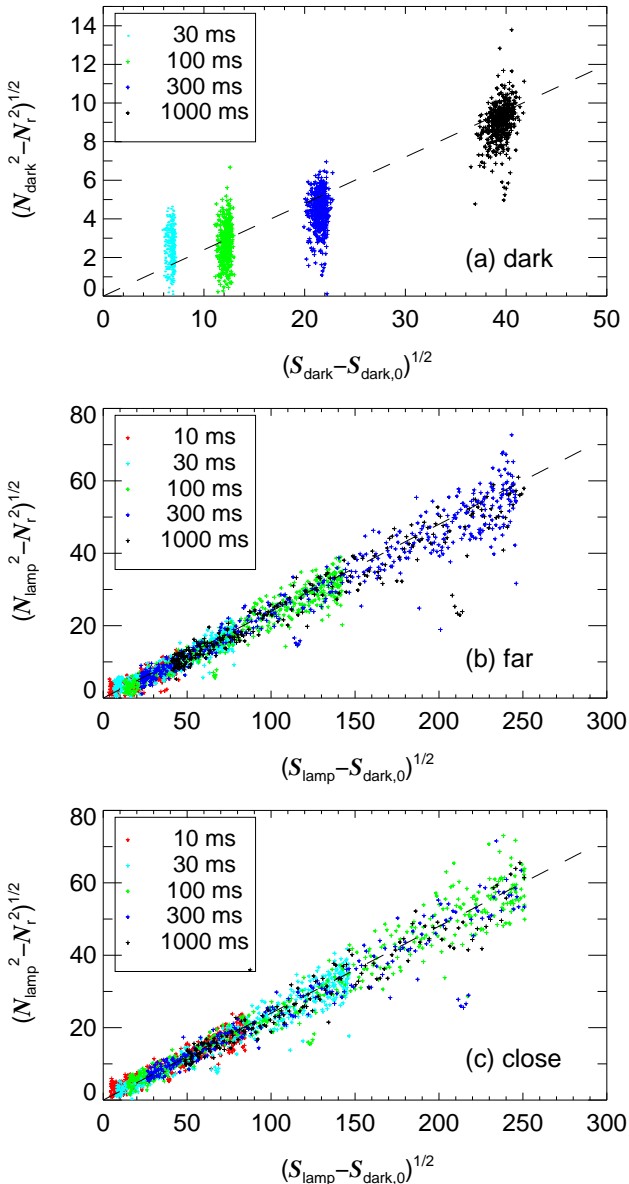

**Figure 4.** Increase of noise ($N$) with the square root of CCD signals from dark measurements ($S_{\mathrm{dark}}$) and 1000 W irradiance standard measurements ($S_{\mathrm{lamp}}$) of instrument 62001: (a) dark, (b) lamp distance 700 mm (far), and (c) lamp distance 350 mm (close). Noise was corrected for residual noise ($N_r \approx 7$) at zero integration time and signals were corrected for constant electronic offsets ($S_{\mathrm{dark},0} \approx 900$). For the dark measurements at integration times $\leq 30$ ms the noise increase is too small to be determined correctly. The dashed lines indicate an approximate linear dependence.

The increase of instrument noise upon exposure to radiation was investigated by measurements with a 1000 W halogen lamp, operated with a highly stabilized power supply in the laboratory. The lamp was providing constant irradiation conditions that also served for spectral calibrations as described in the next section. The lamp was located at two distances from the optical receivers: at the certified distance for absolute calibrations of 700 mm and a smaller distance of about 350 mm. The smaller distance was mainly used to increase signals in the UV-B range. Total noise $N_{\mathrm{lamp}}$ was derived from standard deviations of repeated measurements under the various signal levels $S_{\mathrm{lamp}}$ produced by the lamp. Noise levels were found to increase with the square root of signals and integration times, respectively, consistent with additional photo-electron induced shot noise ($N_{\mathrm{s}}$) as shown in panels (b) and (c) of Fig. 4. This indicates that the noise of the lamp output is insignificant which is in line with the certified $\pm 10$ ppm current stability of the power supply. Thus, for the total noise the following equations apply:

$$N_{\mathrm{lamp}}^2 \quad \approx \quad N_{\mathrm{s}}^2 + N_{\mathrm{d}}^2 + N_{\mathrm{r}}^2 \tag{6}$$

$$\sqrt{(N_{\mathrm{s}}^2 + N_{\mathrm{d}}^2)} \quad \propto \quad \sqrt{(S_{\mathrm{lamp}} - S_{\mathrm{dark},0})}$$

The proportionality factor $\sqrt{(1/G)} \approx 0.25$ indicated in Fig. 4 is consistent with an inverse gain $G \approx 15$ ($\mathrm{e^- ADU^{-1}}$) derived from CCD manufacturer information. Equation 6 is also valid for atmospheric measurements and will be used to calculate the noise of simulated and measured atmospheric signals in Sects. 2.2.7 and 3.3.

Signal-to-noise ratios (SNR) of instrument 62001 as a function of wavelength and integration time are shown in Fig. 5 for the two selected lamp distances. Because only signals induced by desired radiation are usable, mean dark signals and contributions from stray light ($S_{\mathrm{stray}}$) (Sect. 2.2.3) were subtracted in the SNR calculations:

$$\mathrm{SNR} = \frac{S_{\mathrm{lamp}} - S_{\mathrm{dark}} - S_{\mathrm{stray}}}{N_{\mathrm{lamp}}} \tag{7}$$

For a given integration time, the SNR drops strongly towards short wavelengths because of decreasing lamp output, but also because of a decreasing spectral sensitivity (Sect. 2.2.3). On the other hand, for a given wavelength, the SNR increases with integration time and also improves at the shorter lamp distance unless saturation is reached. All other instruments showed a comparable behaviour with the SNR reaching a maximum of around 1000 close to saturation levels. In the following section the advantages of long integration times and short lamp distances are utilized to optimize the calibration procedure.

### 2.2.3 Radiometric laboratory calibration

To calibrate the spectroradiometers, a PTB (Physikalisch-Technische Bundesanstalt) traceable spectral irradiance standard (Gigahertz-Optik, BN-9101) and a suitable power supply (Opteema, OL83A) were used, utilizing the fact that irradiance and actinic flux density are identical upon normal incidence. However, for a point source like a lamp the certified distance between the lamp and the receiver has to be strictly adhered to, in this case 700 mm. For flat irradiance receivers this is straightforward but actinic radiation receivers are composed of quartz domes with an outer vertical extension of about 35 mm with no obvious reference plane. We therefore adapted the concept of equivalent plane receivers (EPR) described in detail by Hofzumahaus et al. (1999). Basically each actinic receiver was characterized by distance dependent measurements of the lamp signal to evaluate

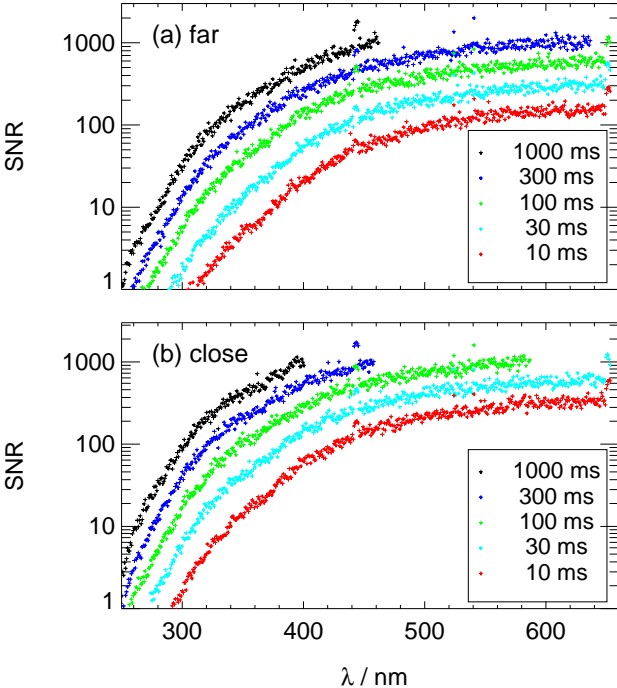

**Figure 5.** Signal-to-noise ratios (SNR) of 1000 W irradiance standard measurements with instrument 62001 as a function of wavelength for different integration times according to Eq. 7: (a) lamp distance 700 mm, (b) lamp distance 350 mm. The SNR is determined by the spectral lamp output and the instrument's spectral sensitivity that together produce the signal height and the corresponding noise (Fig. 4). The improvements for the close-measurements are most useful for wavelengths below 400 nm.

the position of the EPR plane that is typically located 15–25 mm below the quartz dome tip and shows little wavelength dependence (<2 mm, 300–650 nm).

For calibrations, lamp and receiver were mounted on an optical bench at the reference distance between the lamp and the quartz dome tip with the help of a 700 mm spacer. Using the scale of the optical bench the receiver was then moved towards the lamp by the receiver specific EPR plane distance. During calibration measurements the receiver was placed into a black box where the lamp radiation entered through a blind. The blind could be blocked for dark measurements. Alternatively a filter holder with a cutoff filter could be placed in front of the blind. The applied cutoff filter was a WG320 long pass filter (Schott, White Glass) with an edge wavelength of 320 nm that safely removes all radiation below 300 nm (<1%) for separate stray light measurements in that range. Occasionally, further cutoff filters were used, in particular a WG360 with a 360 nm edge wavelength.

A typical calibration was made by a sequence of four cycles of 50–100 single measurements each, comprising different integration times of up to 1000 ms: (1) dark, (2) lamp, (3) filter, and (4) dark. The receiver was then moved by 300-400 mm towards the lamp and the procedure was repeated. These cycles are referred to as far- and close-measurements. While for the far-measurements the correct distance to the lamp is important, the distance of the close-measurements is secondary, as long

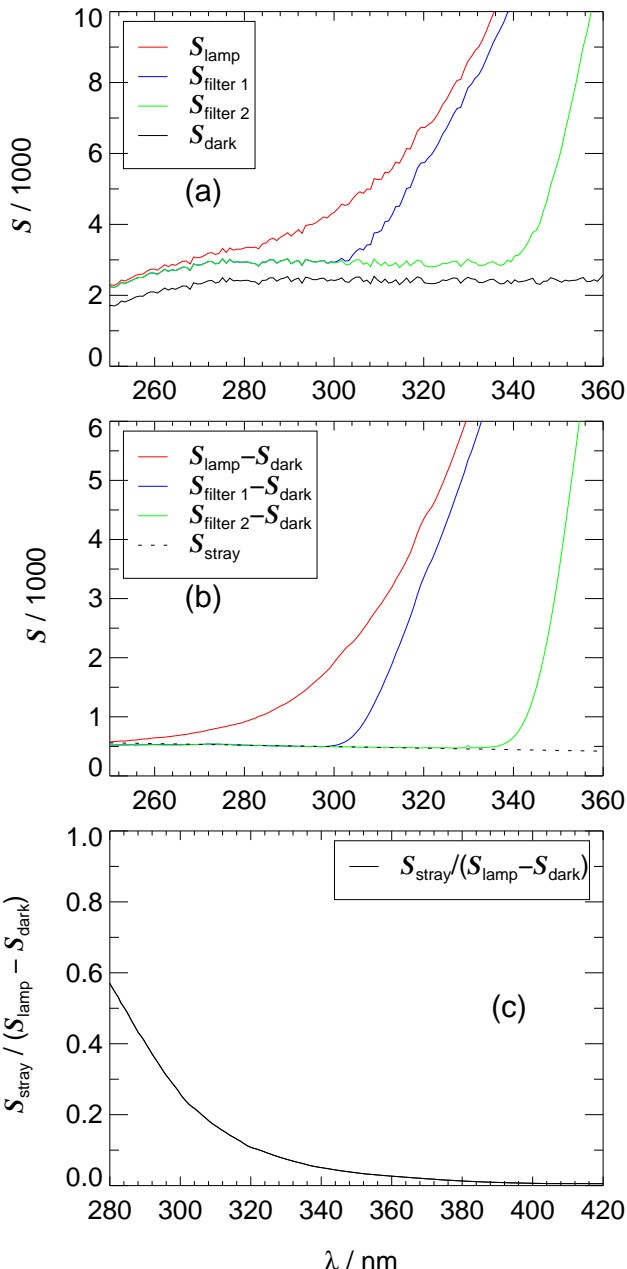

**Figure 6.** Example signals obtained during laboratory calibrations of instrument 62001 with two cutoff filters at $\Delta t$=1000 ms and close lamp distance. The wavelength range relevant for the determination of stray light signals is zoomed in. Saturation occurred around 400 nm for this integration time. (a) Total signals of lamp radiation with no filter ($S_{\mathrm{lamp}}$), of lamp radiation with WG320 filter ($S_{\mathrm{filter\ 1}}$), of lamp radiation with WG360 filter ($S_{\mathrm{filter\ 2}}$), and dark signals ($S_{\mathrm{dark}}$). (b) After subtraction of dark signals, stray light signals were estimated by linear regressions in a range 270-300 nm and extrapolated over the whole spectral range (dashed lines) before final subtraction. (c) Signal contribution of stray light in the atmospherically most relevant wavelength range. Similar figures for the other instruments are shown in the Supplement.

as a substantial increase of signals by a factor 3-5 is achieved. On the other hand, it is crucial that accurate dark signals are determined and subtracted. The dark measurements before and after the lamp measurements were routinely made to allow for a check that there was no significant drift in dark signals during the calibration measurements.

The method to improve the calibration accuracy by using two lamp distances was recommended by the manufacturer and was already applied for diode array based spectroradiometers (PDA-SR) (Kanaya et al., 2003; Edwards and Monks, 2003; Jäkel et al., 2005; Bohn et al., 2008) and CCD-SR (Jäkel et al., 2007). The procedures described in the following were elaborated to improve the determination of stray light signals and to utilize several integration times to obtain optimized spectral sensitivities.

Spectral sensitivities $D_\lambda$ were calculated using the following equation:

$$D_\lambda(\lambda, \Delta t) = \frac{S_{\text{lamp,corr}}^{\text{close}}(\lambda, \Delta t)}{E_\lambda^{\text{std}}(\lambda) \times f_1} \tag{8}$$

$S_{\text{lamp,corr}}^{\text{close}}(\lambda, \Delta t)$ are spectroradiometer signals from the close-measurements at different integration times $\Delta t$, corrected for dark and stray light signals. $E_\lambda^{\text{std}}(\lambda)$ are the certified spectral irradiances of the standard lamp in the required spectral flux density units ($\text{cm}^{-2}\text{s}^{-1}\text{nm}^{-1}$) and $f_1$ is the mean ratio of corrected signals from close- and far-measurements. These quantities will be explained in more detail in the following. Note that all $S$ variables are averages that depend on integration time and wavelength which will not be indicated explicitly in the following equations for brevity.

Close-measurement signals ($S_{\text{lamp}}^{\text{close}}$) were corrected by subtraction of dark and estimated stray light signals ($S_{\text{stray}}^{\text{close}}$) corrected by a further scaling factor $f_2^{\text{close}}$:

$$S_{\text{lamp,corr}}^{\text{close}} = S_{\text{lamp}}^{\text{close}} - S_{\text{dark}} - S_{\text{stray}}^{\text{close}} \times f_2^{\text{close}} \tag{9}$$

Figure 6 shows examples of signals from close-measurements for an integration time of 1000 ms. In panel (a), dark signals, lamp signals without filter and lamp signals with two different cutoff filters (WG320, WG360) ($S_{\text{filter}}^{\text{close}}$) are plotted. For better visibility the wavelength range was confined to 250–360 nm. In panel (b), dark signals were subtracted which also removes the dark current induced fluctuations, as intended (Sect. 2.2.2). The remaining signals come from the desired lamp radiation and underlying stray light. Obviously the $S_{\text{stray}}^{\text{close}}$ can only be determined from measurements with cutoff filter in a wavelength range where the filter safely blocks radiation, i.e. below 300 nm in the case of a WG320 filter:

$$S_{\text{stray}}^{\text{close}} = S_{\text{filter}}^{\text{close}} - S_{\text{dark}} \ (< 300 \ \text{nm}) \tag{10}$$

For all instruments $S_{\text{stray}}^{\text{close}}$ could be linearly fitted in good approximation in a range around 270–300 nm. The $S_{\text{stray}}^{\text{close}}$ at greater wavelengths were then approximated by linear extrapolations over the full spectral range, as indicated by the dashed line in panel (b) of Fig. 6. The validity of this extrapolated stray light signal is confirmed in this example by the additional measurements with the WG360 filter showing an almost identical stray light background compared to the extrapolation.

In panel (c) of Fig. 6 the fraction of stray light signals is plotted for the atmospherically most relevant wavelength range 280–420 nm. With increasing wavelength and lamp signals, the importance of stray light quickly diminishes to below 5% above 350 nm. Accordingly, uncertainties of the extrapolations become unimportant. Linear extrapolations of stray light signals were preferred in this work because they were most accurate in a range $<350$ nm where the determination of $D_\lambda$ is strongly affected

by stray light. A modification of this procedure was only necessary for the oldest instrument 45853 where a stronger wavelength dependence and a leveling-off of the stray light induced signal around 340 nm was observed. For 45853 the stray light level was generally increased compared to the other instruments.

Plots as in Fig. 6 can be found for all instruments in the Supplement using the same typical optical receiver/fiber combination for direct comparison. If for other instruments a linear fit or a linear extrapolation of stray light signals towards longer wavelengths turns out to be insufficient, other functions should be tested to obtain an optimum description. For example, for 45853 a second-order polynomial was used for the extrapolation of stray light signals. This polynomial had the same slope at 300 nm as the linear approximation (270-300 nm) but was allowed to smoothly level out around 340 nm. Further cutoff filters with longer edge wavelengths were used to exclude significant spectral structures in the stray light signal at longer wavelengths. However, the spectral shape of the stray light signal without filter may differ from that observed with a filter below its cutoff wavelength because the filter can remove a substantial part of the radiation responsible for stray light. For that reason it is important to use filters with short cutoff wavelengths like a WG320 to get reliable results in the most affected wavelength range.

The further scaling factor $f_2^{\text{close}}$ in Eq. 9 accounts for the fact that the WG320 filter slightly diminishes radiation also well above its cutoff wavelength by normal reflections at the filter surfaces. Accordingly, stray light is slightly smaller during the filter measurements. Because a large fraction of stray light ($\approx$50%) originates in the unmeasured VIS and NIR range of the spectrum, as was verified by cutoff filters with longer edge wavelengths, $f_2^{\text{close}}$ was determined in a range 640$\pm$10 nm.

$$f_2^{\text{close}} = \frac{S_{\text{lamp}}^{\text{close}} - S_{\text{dark}}}{S_{\text{filter}}^{\text{close}} - S_{\text{dark}}} \; (640 \pm 10 \text{ nm}) \tag{11}$$

The $f_2^{\text{close}}$ typically ranged around 1.05 and were found to be independent of integration time, as expected. Consequently, mean values of $f_2^{\text{close}}$ were applied using integration times not affected by saturation. For the spectral sensitivities the factor $f_2^{\text{close}}$ is negligible above 310 nm ($<$1%) but it becomes increasingly important at shorter wavelengths ($\approx$10% at 280 nm). An exception is again instrument 45853 with a greater influence of $f_2^{\text{close}}$ because of an increased stray light level.

Finally the factor $f_1$ in Eq. 8 was determined from mean ratios of corrected close- and far-signals:

$$f_1 = \overline{\frac{S_{\text{lamp,corr}}^{\text{close}}}{S_{\text{lamp,corr}}^{\text{far}}}} \tag{12}$$

As long as the denominator and the numerator in Eq. 12 were below saturation and above a certain, noise insensitive threshold ($\approx$200), respectively, these factors were found to be independent of wavelength and integration times. The scaling factor $f_1$ is important because it establishes the final connection between the SNR-improved close-measurements and the far-measurements at the correct lamp distance.

The procedure described so far yields spectral sensitivities for each integration time that was used during the calibrations. These sensitivities are expected to scale with the integration times:

$$D_\lambda(\lambda, \Delta t_2) = D_\lambda(\lambda, \Delta t_1) \times \frac{\Delta t_2}{\Delta t_1} \tag{13}$$

The relation in Eq. 13 was tested by comparing measured sensitivities for an integration time of 10 ms and calculated sensitivities for the same integration time from measurements at 30, 100, 300 and 1000 ms as shown in Fig. 7. Sensitivities at

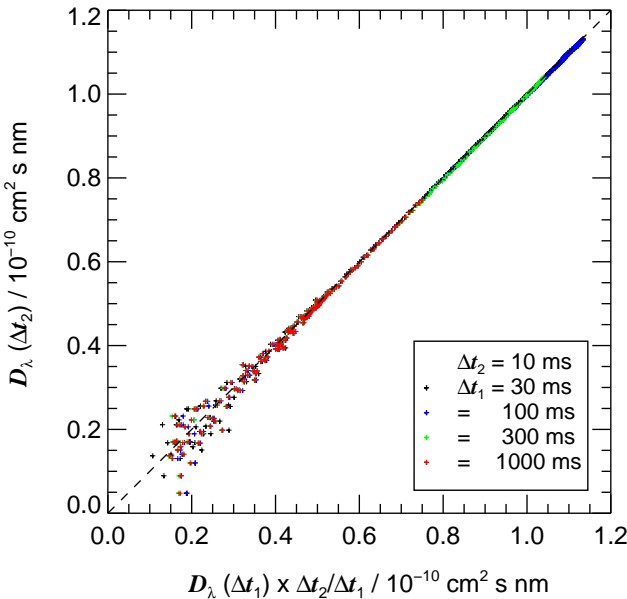

**Figure 7.** Comparison of spectral sensitivities of instrument 62001 for an integration time of 10 ms ($\Delta t_2$) with scaled spectral sensitivities obtained at other integration times ($\Delta t_1$). Scaling factors were calculated according to Eq. 13. Data points cover all wavelengths. The dashed line indicates perfect linearity.

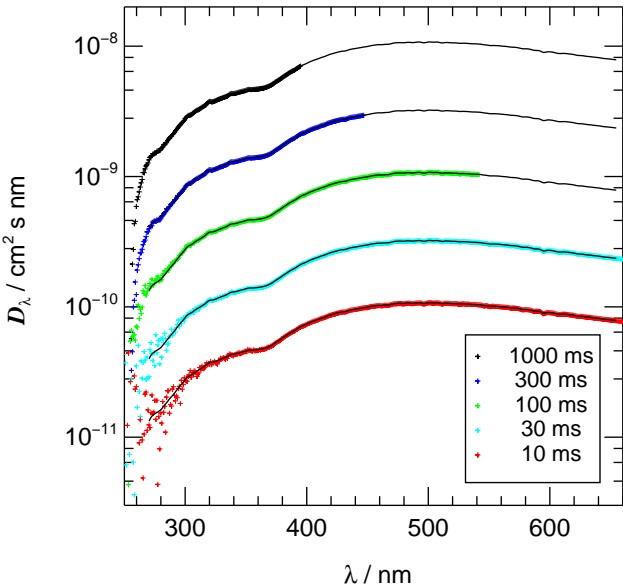

**Figure 8.** Spectral sensitivities of instrument 62001 at different integration times. Full lines show an optimized sensitivity that was scaled dependent on integration time according to Eq. 13. Data points show the separately measured sensitivities for comparison.

10 ms were chosen as an example because no saturation occurred, i.e. the comparison is covering all wavelengths (but not at all integration times). Linear regressions produced slopes deviating less than 1% from unity confirming the strict linearity of the measurements. Equation 13 was therefore used to derive consistent, optimum sensitivities for all integration times, by favoring measurements with the longest possible $\Delta t_1$ as long as no saturation was reached. Practically, a master sensitivity file was produced for the maximum integration time of 1000 ms. The data were then scaled to obtain sensitivities for other integration times. An example of the resulting sensitivities is shown in Fig. 8 (full lines) together with the measured values. With decreasing integration time and wavelengths, measured data expectedly start to scatter around the optimized sensitivities. The common decrease of sensitivities towards shorter wavelengths is, by the way, not caused by a decrease of CCD sensitivity but mainly by the actinic radiation receiver through which only a small fraction of multiply scattered radiation is eventually transmitted. Moreover, long optical fibers have an adverse effect on spectral sensitivities with somewhat stronger attenuations towards shorter wavelengths.

The absolute sensitivities obtained for all instruments are comparable and roughly correspond to that shown by Jäkel et al. (2007) for their original setup. However, for the majority of their measurements Jäkel et al. (2007) employed a UV transmitting filter (UG5, Schott) that strongly diminished the sensitivity above 400 nm. Through this modification they reduced stray light, avoided saturation of the CCD in the VIS range and could therefore work with a single integration time of 200 ms during field measurements. However, the stray light reduction by a UG5 is only about 50% because this filter still transmits a substantial fraction of NIR radiation. At the same time the UG5 transmittance is only 1% around 600 nm which could introduce a stray light issue in a spectral range important for $NO_3$ photolysis. In contrast to the UV-B range where the atmospheric cut-off provides a means to routinely determine the variable stray light contribution during field measurements (Sec. 3.1), there is no such possibility in the VIS range. We therefore accept the inconvenience of multiple integration times and of increased stray light to exclude any interference outside the UV-B. Nevertheless, the approach by Jäkel et al. (2007) is generally supported for atmospheric measurements, unless small values of $j(NO_3)$ are of interest. Jäkel et al. (2007) also compared the performance of PDA-SR and CCD-SR in the UV range and clearly demonstrated the advantage of CCD-SR because of their higher sensitivity. Also Eckstein et al. (2003) described a CCD-SR with similar sensitivities and a time resolution of 3 s. A direct comparison of spectral sensitivities is difficult because a teflon sphere was used for $4\pi$ measurements of spectral actinic flux densities with a single receiver.

### 2.2.4 In-field calibrations

For technical reasons, in-field calibrations with a 1000 W standard (as described in the previous section) are difficult, especially for the final setup on an aircraft. On the other hand, calibrations are necessary to monitor any sensitivity change caused by transportation or the installation process which usually requires that optical fibers are disconnected, rearranged and reconnected. Therefore, secondary calibrations with small 45 W lamps acting as traveling standards were made. These lamps have specially designed housings that can be fixed directly at the optical receivers without any interference to receiver mountings or optical fibers. Moreover, the ventilated lamps are shielded against ambient radiation so that calibrations are feasible during daylight. Figure 9 shows the setup during a calibration in the HALO hangar. Spectra of two 45 W lamps were routinely recorded directly

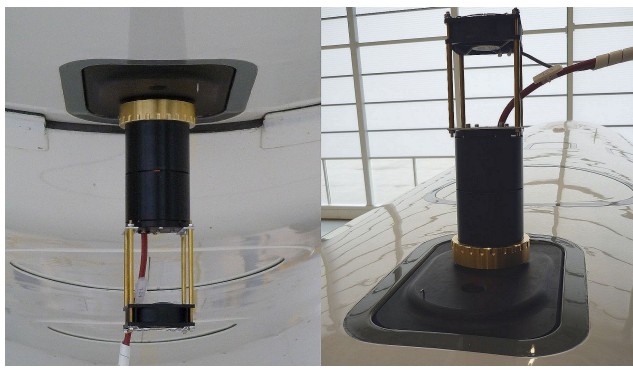

**Figure 9.** In-field calibration setup with traveling standard lamps attached to receiver optics on HALO. Left: bottom fuselage, right: top fuselage.

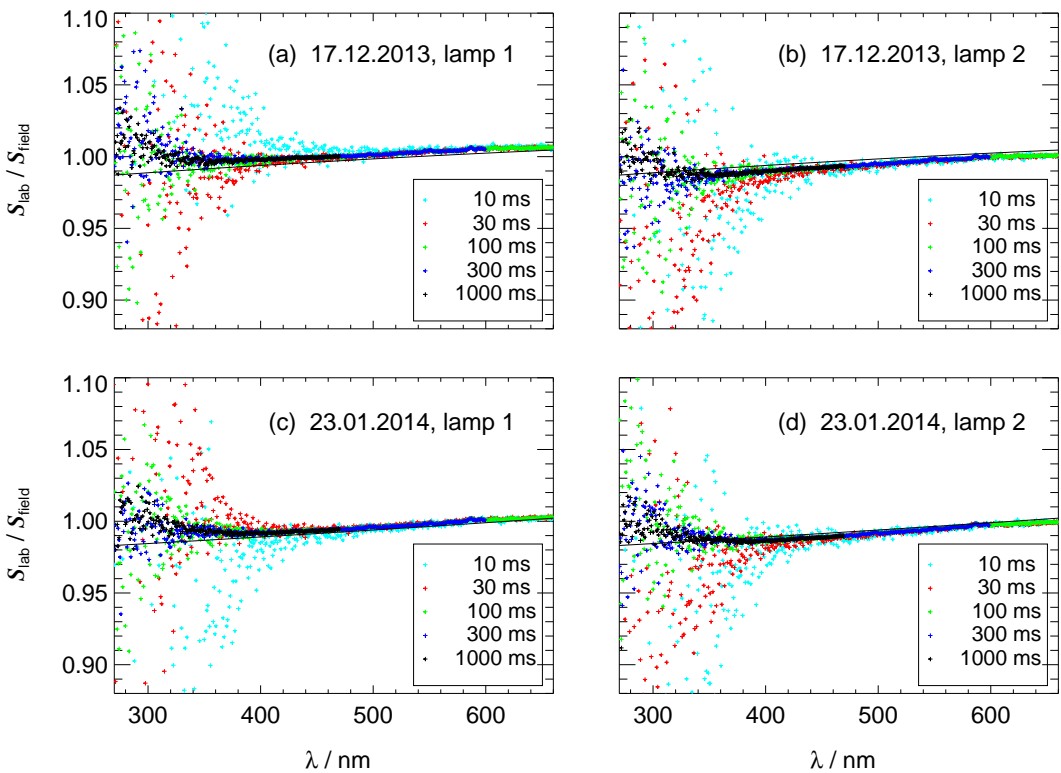

**Figure 10.** Ratios of dark-corrected signals of 45 W traveling standard lamps obtained with instrument 62001 after a laboratory calibration on 03.09.2013 ($S_{lab}$) and directly before and after a deployment on HALO ($S_{field}$) in panels (a, b) and (c, d), respectively. Two lamps were used each during laboratory and field measurements. Data points show individual measurements for different integration times, full lines the finally applied scaling factors for each date (polynomial fits, mean of measurements with both lamps on a specific date).

after the laboratory calibration with the irradiance standard, as well as in the aircraft hangars before and after the instrument deployments. The use of two lamps allows for consistency checks and assures that the transfer calibration is not lost in case of a lamp failure.

Figure 10 shows examples of measurements that were made with instrument 62001 before and after a deployment on HALO. Ratios of dark signal corrected lamp signals are shown representing the relative change of spectral sensitivities compared to the laboratory calibration. As in the laboratory, 100 single dark and lamp measurements were always averaged. The large scatter of the ratios at short integration times and below 350 nm results from the low output of the 45 W lamps that is comparable to far-measurements with a 1000 W lamp. Moreover, for technical reasons the optional use of cutoff filters was not feasible for the small lamps. As a result, the change of spectral sensitivities in the UV-B cannot be determined accurately. On the other hand, little, if any spectral variations have been observed above 350 nm. Therefore, extrapolations of second-order polynomials were used below 350 nm that were fitted in a range 350–650 nm. The full lines in Fig. 10 indicate the corresponding ratios that consider the measurements with both lamps and that were finally applied to scale the laboratory based spectral sensitivities. Typically the scaling factors ranged between 0.95 and 1.05 which means that laboratory calibrations were widely reproducible after transportation and aircraft installation. Moreover, measurements before and after deployments were typically within 2%, i.e. calibrations were stable as long as the setups remained unchanged. Finally, the results obtained with two different lamps were usually similar within 1%, giving additional confidence in the field-calibration procedure.

### 2.2.5 Spectral calibration accuracy

The overall accuracy of the spectral calibrations is determined by a number of factors; firstly by the certified accuracy of the irradiance standard which is 3–4%, dependent on wavelength. The accuracy of the lamp current produced by the power supply is certified with 0.01% which translates to a maximum 0.1% uncertainty of the irradiance output (manufacturer information). A further 1.5% uncertainty is calculated from an estimated 5 mm uncertainty of the position of the EPR reference plane. Because consistent results were obtained for different wavelengths and integration times, the accuracy of the factor $f_1$ is within 0.5%.

The uncertainties related with the subtracted stray light signals are more difficult to assess. In the most sensitive range below 300 nm the applied linear approximation is leading to deviations less than 1% from the measured values. The uncertainty of the extrapolation beyond 300 nm is increasing with wavelength but the importance of stray light also strongly decreases with increasing wavelength. Moreover, an additional 1% uncertainty is estimated for the scaling factor $f_2$ of the stray light signal. Assuming a total 3% uncertainty of the subtracted stray light signals, changes in sensitivities between 1% at 300 nm and 0.02% around 400 nm are obtained (4% and 0.1% for instrument 45853). Finally, the accuracy of in-field calibrations is estimated 2% in the UV-B and 1% for the UV-A and VIS range. Taking all these factors together results in total spectral calibration uncertainties between 5–6% at 300 nm and 4% at 650 nm.

These uncertainty estimates were derived from carefully controlled laboratory measurements at normal incidence of radiation and roughly apply for any CCD spectroradiometer with similar properties. However, total atmospheric measurement uncertainties can be affected by additional factors related with receiver specific angular response imperfections, atmospheric stray light influence and instrument noise that are dependent on measurement conditions. Based on the laboratory characteri-

zations, the influence of instrument noise on detection limits and measurement precisions is estimated in the following before atmospheric measurements are addressed in Sect. 3.

### 2.2.6 Detection limits and cutoff wavelengths

During laboratory calibrations measurements were repeated to reduce the noise. For example, averaging over 100 single measurements reduces the noise by a factor of $\sqrt{100}$. Instrument noise therefore plays no important role for the determination of spectral sensitivities. Also during field measurements, averaging is possible and often applied. However, averaging also leads to a reduction of time resolution which may not be useful for airborne measurements. Therefore, at maximum time resolution the noise of single measurements determines precisions and detection limits. In order to estimate these limits, a noise-equivalent spectral actinic flux density ($F_{\lambda,\mathrm{dark}}^{\mathrm{NE}}$) can be defined by the ratios of the dark noise obtained in Sect. 2.2.2 and spectral sensitivities:

$$F_{\lambda,\mathrm{dark}}^{\mathrm{NE}}(\lambda,\Delta t) = N_{\mathrm{dark}}(\lambda,\Delta t)/D_\lambda(\lambda,\Delta t) \tag{14}$$

The corresponding spectra for instrument 62001 are shown in Fig. 11 as an example. Expectedly, the $F_{\lambda,\mathrm{dark}}^{\mathrm{NE}}$ increase with decreasing wavelength and are lower for longer integration times. Absolute values are comparable with results obtained by Jäkel et al. (2007) in the UV range but smaller in the VIS range because no UG5 filter was used in this work. Detection limits are usually estimated as the three-fold of the noise-equivalent values (Magnusson and Örnemark, 2014) which can be further improved by a factor $\sqrt{n}$ upon averaging over $n$ single measurements. Without averaging, detection limits $\approx 1 \times 10^{10} \mathrm{cm}^{-2}\mathrm{s}^{-1}\mathrm{nm}^{-1}$ are obtained for wavelengths around 300 nm at 1000 ms integration time. However, these detection limits should be considered as a theoretical minimum derived from dark noise because additional uncertainties from stray light and varying background under field conditions are not included.

Corresponding noise estimates in terms of photolysis frequencies were obtained by multiplying the noise-equivalent spectra with random noise for each pixel followed by calculations of photolysis frequencies according to Eqs. 2 and 3 as an example. The resulting standard deviations for $j(\mathrm{O}^1\mathrm{D})$ and $j(\mathrm{NO}_2)$ for instrument 62001 are listed in Tab. 3 for different integration times. In these calculations the wavelength range was confined to the atmospherically relevant range above 280 nm. For the longest integration time of 1000 ms the noise limits correspond to 0.1% and 0.0001% of typical maximum values of $j(\mathrm{O}^1\mathrm{D})$ ($\approx 4 \times 10^{-5}\mathrm{s}^{-1}$) and $j(\mathrm{NO}_2)$ ($\approx 1 \times 10^{-2}\mathrm{s}^{-1}$), respectively. For the shortest integration time of 10 ms these numbers increase by factors of 40–50, still sufficiently low for $j(\mathrm{NO}_2)$, but not for $j(\mathrm{O}^1\mathrm{D})$. Also for the photolysis frequencies a factor of three should be applied to estimate detection limits which are again regarded as theoretical, dark noise derived minima unless averaging is permitted.

The detection limits of $j(\mathrm{O}^1\mathrm{D})$ can be significantly reduced by further confining the wavelength range. This is demonstrated in Tab. 3 for a wavelength of 300 nm as an example. The reason for this reduction is that $\mathrm{O}(^1\mathrm{D})$ formation mainly takes place in a range below 320 nm where the corresponding product $\sigma \times \phi$ in Eq. 2 increases strongly towards shorter wavelengths. For $j(\mathrm{NO}_2)$ no such reduction of detection limits is obtained because the term $\sigma \times \phi$ is distributed over a wider wavelength range, covering the complete UV with a broad peak around 380 nm.

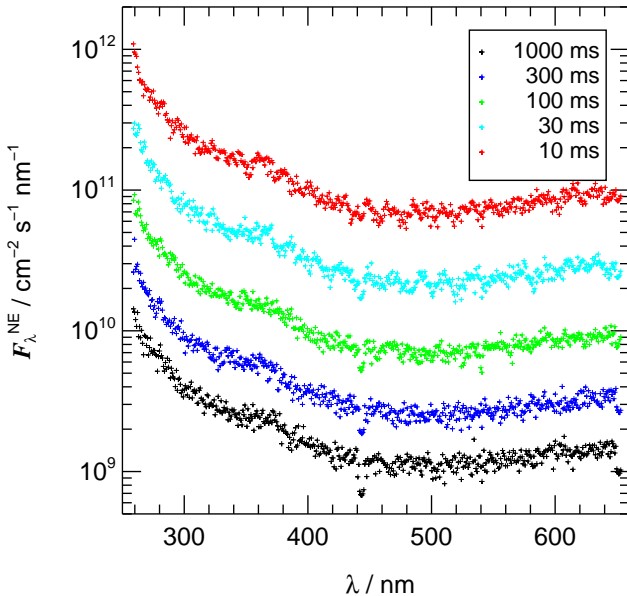

**Figure 11.** Dark noise-equivalent actinic flux density $F_{\lambda,\text{dark}}^{\text{NE}}$ of instrument 62001 for single measurements at different integration times according to Eq. 14.

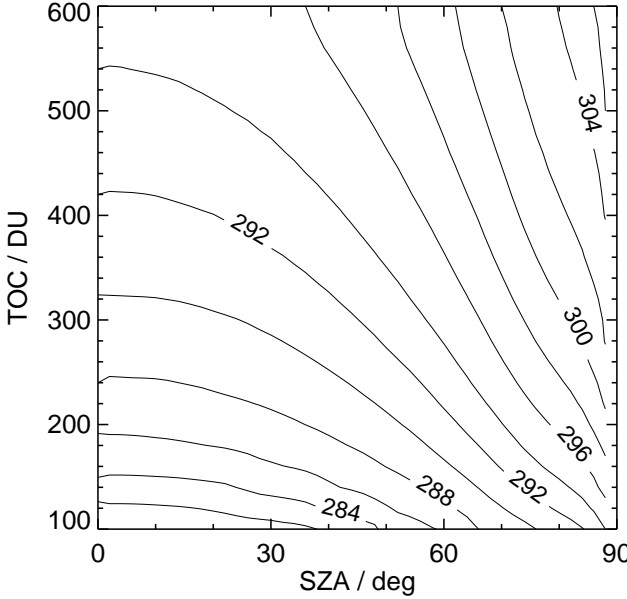

**Figure 12.** Contour plot of atmospheric cutoff wavelengths (nm) for an altitude of 15 km as a function of solar zenith angles (SZA) and total ozone columns (TOC). The data were derived from radiative transfer calculations of downward clear sky spectral actinic flux densities defining a lower limit $F_\lambda \leq 5 \times 10^9 \text{cm}^{-2}\text{s}^{-1}\text{nm}^{-1}$. A similar plot for an altitude of 0 km can be found in the Supplement.

**Table 3.** Standard deviations of photolysis frequencies obtained with random noise corresponding to dark noise-equivalent actinic flux densities (Eq. 14) for instrument 62001. The spectral range is 280–650 nm, for numbers in brackets 300–650 nm.

| integration-time / ms | dark noise equivalent photolysis frequency / s$^{-1}$ | |
| :---: | :---: | :---: |
| | $j(O^1D)$ | $j(NO_2)$ |
| 10 | $2.6{\times}10^{-6}$ ($1.3{\times}10^{-7}$) | $5.9{\times}10^{-7}$ ($5.8{\times}10^{-7}$) |
| 30 | $8.4{\times}10^{-7}$ ($4.6{\times}10^{-8}$) | $2.0{\times}10^{-7}$ ($1.9{\times}10^{-7}$) |
| 100 | $2.6{\times}10^{-7}$ ($1.4{\times}10^{-8}$) | $6.1{\times}10^{-8}$ ($6.0{\times}10^{-8}$) |
| 300 | $9.9{\times}10^{-8}$ ($5.4{\times}10^{-9}$) | $2.4{\times}10^{-8}$ ($2.3{\times}10^{-8}$) |
| 1000 | $3.9{\times}10^{-8}$ ($2.2{\times}10^{-9}$) | $1.2{\times}10^{-8}$ ($9.4{\times}10^{-9}$) |

Because of the strong spectral weighting in the range 280–320 nm, improvements of the $j(O^1D)$ noise or measurement precision can be achieved under all atmospheric conditions by confining the wavelength range. Of course, to what extend such a confinement is justified depends on measurement conditions because the wavelength below which atmospheric actinic flux density can be safely set to zero because it becomes negligible in terms of $j(O^1D)$ is mainly determined by total ozone columns

(TOC) and solar zenith angles (SZA). In the following we define so-called cutoff-wavelengths below which spectral actinic flux densities safely drop below values of $5{\times}10^9 \text{cm}^{-2}\text{s}^{-1}\text{nm}^{-1}$ which roughly corresponds to the $F_{\lambda,\text{dark}}^{\text{NE}}$ around 300 nm for the longest integration times of the spectroradiometers (Fig. 11). Practically, the cutoff wavelengths were derived from model calculations of clear-sky downward spectral actinic flux densities using the libRadtran radiative transfer model (Mayer and Kylling, 2005; Emde et al., 2016). Calculations were made for altitudes of 0 km (ground-based and Zeppelin measurements)

and 15 km (aircraft measurements) covering TOC=100–600 DU (Dobson units) and SZA=0–88°. For both altitudes lookup tables were produced ranging between 280 nm (15 km, SZA=0°, TOC=100 DU) and 309 nm (0 km, SZA=88°, TOC=600 DU). A contour plot of 15 km cutoff wavelengths is shown in Fig. 12 for illustration, a corresponding plot for 0 km can be found in the Supplement. Typically the differences between 0 km and 15 km cutoff wavelengths are no more than around 2 nm and the $j(O^1D)$ fractions attributable to the wavelength ranges below the cutoffs are always insignificant (<0.1%).

The cutoff wavelengths were not only introduced here to improve the precision of $j(O^1D)$ measurements but also to determine the variable wavelength limits below which atmospheric stray light signals can be determined in a similar way as during laboratory measurements with cutoff filters. This approach will be applied and explained in more detail in Sect. 3.1.2.

### 2.2.7 Measurement precisions

The influence of radiation induced shot noise on the precision of $F_\lambda$ measurements as well as the effect of cutoff wavelengths

on the precision of $j(O^1D)$ and $j(NO_2)$ under various atmospheric conditions were investigated based on the same simulated clear-sky downward $F_\lambda$ spectra that were used to derive the cutoff wavelengths. Signal spectra for different integration times were calculated from the $F_\lambda$ by multiplication with the spectral sensitivities of instrument 62001 (Fig. 8). After addition of

**Table 4.** Downward clear-sky spectral actinic flux densities $F_\lambda$ from radiative transfer calculations for selected wavelengths and solar zenith angles (SZA) for an altitude of 15 km and an ozone column of 300 DU (left) and simulated noise equivalent actinic flux densities $F_\lambda^{\mathrm{NE}}$ of instrument 62001 for a maximum 300 ms integration time (right). The entry SZA>100° indicates dark conditions.

| $\lambda$ / nm | 300 | 350 | 400 | 450 | 500 | 550 | 600 | 650 | 300 | 350 | 400 | 450 | 500 | 550 | 600 | 650 |
|---|---|---|---|---|---|---|---|---|---|---|---|---|---|---|---|---|
| SZA / deg | $F_\lambda$ / $10^{12}\mathrm{cm}^{-2}\mathrm{s}^{-1}\mathrm{nm}^{-1}$ | | | | | | | | $F_\lambda^{\mathrm{NE}}$ / $10^{10}\mathrm{cm}^{-2}\mathrm{s}^{-1}\mathrm{nm}^{-1}$ | | | | | | | |
| 0 | 5.8 | 200 | 360 | 500 | 490 | 520 | 530 | 510 | 2.2 | 29 | 52 | 53 | 50 | 54 | 58 | 61 |
| 30 | 4.0 | 200 | 360 | 500 | 490 | 510 | 520 | 510 | 1.9 | 28 | 52 | 53 | 50 | 54 | 57 | 61 |
| 50 | 1.5 | 200 | 360 | 490 | 480 | 510 | 520 | 510 | 1.4 | 28 | 52 | 52 | 50 | 53 | 57 | 60 |
| 60 | 0.50 | 190 | 350 | 490 | 480 | 500 | 510 | 500 | 1.1 | 28 | 52 | 52 | 50 | 53 | 57 | 60 |
| 70 | 0.06 | 180 | 340 | 470 | 470 | 490 | 490 | 490 | 1.0 | 27 | 51 | 51 | 49 | 52 | 55 | 59 |
| 80 | $0.55^a$ | 140 | 290 | 430 | 430 | 440 | 430 | 460 | 1.0 | 24 | 47 | 49 | 47 | 50 | 52 | 57 |
| 84 | $0.32^a$ | 100 | 240 | 380 | 380 | 380 | 360 | 410 | 1.0 | 11 | 24 | 46 | 45 | 46 | 48 | 55 |
| 88 | $0.14^a$ | 26 | 93 | 200 | 210 | 190 | 160 | 250 | 1.0 | 3.2 | 15 | 19 | 34 | 19 | 18 | 25 |
| >100 | 0 | 0 | 0 | 0 | 0 | 0 | 0 | 0 | 1.0 | 0.6 | 0.4 | 0.3 | 0.3 | 0.3 | 0.3 | 0.4 |

$^a$ $F_\lambda$ / $10^{10}\mathrm{cm}^{-2}\mathrm{s}^{-1}\mathrm{nm}^{-1}$

**Table 5.** Photolysis frequencies calculated from the actinic flux density spectra of Tab. 4 (left) and simulated noise equivalent photolysis frequencies of instrument 62001 for a maximum 300 ms integration time (right). $j(\mathrm{O^1D})$ precisions in brackets were obtained by applying variable cutoff wavelengths (Fig. 12). The entry SZA>100° indicates dark conditions with zero spectral actinic flux densities.

| SZA / deg | photolysis frequency | | noise equivalent photolysis frequency | |
|---|---|---|---|---|
| | $j(\mathrm{O^1D})$ / s$^{-1}$ | $j(\mathrm{NO_2})$ / s$^{-1}$ | $j(\mathrm{O^1D})$ / s$^{-1}$ | $j(\mathrm{NO_2})$ / s$^{-1}$ |
| 0 | $6.09\times10^{-5}$ | $9.56\times10^{-3}$ | $1.1\times10^{-7}$ $(3.6\times10^{-8})$ | $1.2\times10^{-6}$ |
| 30 | $4.99\times10^{-5}$ | $9.50\times10^{-3}$ | $1.0\times10^{-7}$ $(3.0\times10^{-8})$ | $1.2\times10^{-6}$ |
| 50 | $3.20\times10^{-5}$ | $9.27\times10^{-3}$ | $1.0\times10^{-7}$ $(2.1\times10^{-8})$ | $1.2\times10^{-6}$ |
| 60 | $2.13\times10^{-5}$ | $8.98\times10^{-3}$ | $1.0\times10^{-7}$ $(1.5\times10^{-8})$ | $1.2\times10^{-6}$ |
| 70 | $1.09\times10^{-5}$ | $8.38\times10^{-3}$ | $1.1\times10^{-7}$ $(9.7\times10^{-9})$ | $1.2\times10^{-6}$ |
| 80 | $3.01\times10^{-6}$ | $6.81\times10^{-3}$ | $1.1\times10^{-7}$ $(5.7\times10^{-9})$ | $9.7\times10^{-7}$ |
| 84 | $1.14\times10^{-6}$ | $5.25\times10^{-3}$ | $1.1\times10^{-7}$ $(4.7\times10^{-9})$ | $7.2\times10^{-7}$ |
| 88 | $1.42\times10^{-7}$ | $1.68\times10^{-3}$ | $1.0\times10^{-7}$ $(3.7\times10^{-9})$ | $2.7\times10^{-7}$ |
| >100 | 0.0 | 0.0 | $1.0\times10^{-7}$ $(3.6\times10^{-9})$ | $2.4\times10^{-8}$ |

mean dark signals the corresponding noise was obtained according to Eq. 6 and optimized noise spectra were combined by preferring long integration times unless saturation levels were reached. A maximum 300 ms integration time was assumed as during atmospheric measurements (Sect. 3.1.2) and the respective noise equivalent $F_\lambda^{\mathrm{NE}}$ were derived. The results are listed

in Tab. 4 together with the $F_\lambda$ from the model for a number of solar zenith angles and wavelengths at an altitude of 15 km. With increasing signals, shot noise increases and shorter integration times become necessary. Accordingly, the $F_\lambda^{\text{NE}}$ increase with wavelength and solar elevation. A comparison of $F_\lambda$ and $F_\lambda^{\text{NE}}$ shows that except for the shortest wavelength, high signal-to-noise ratios ($\geq$600) can be expected under all conditions. For a zero-spectrum the results obtained with the measured dark spectra in Fig. 11 for an integration time of 300 ms were reproduced. The corresponding data for an altitude of 0 km can be found in the Supplement. The potential influence of stray light signals is not considered in this analysis. However, as will be shown in Sect. 3, stray light induced shot noise is very limited.

By repeatedly applying random $F_\lambda^{\text{NE}}$ noise for each pixel, simulated precisions of photolysis frequencies for instrument 62001 were obtained. These data are listed in Tab. 5 for the same conditions as in Tab. 4. For $j(\text{O}^1\text{D})$ the precision is almost constant and independent of the photolysis frequency because radiation induced shot noise is apparently secondary. The application of cutoff wavelengths from the lookup tables led to significant improvements in particular towards large SZA because of increasing cutoff wavelengths. In contrast, for $j(\text{NO}_2)$, photon induced shot noise plays an important role and several shorter integration times were involved in the simulated measurements. Accordingly, the absolute noise increases with increasing $j(\text{NO}_2)$. The application of cutoff wavelengths led to no changes for $j(\text{NO}_2)$, the results were therefore not included in Tab. 5. For a zero-spectrum the results obtained with the measured dark spectra for an integration time of 300 ms were again reproduced (Tab. 3). Corresponding data for an altitude of 0 km can be found in the Supplement.

## 3 Field measurements and data analysis

### 3.1 Field data evaluation

#### 3.1.1 Auxiliary data

In order to simplify the data analysis of aircraft measurements, sets of all required additional parameters were collected in separate files after synchronisation with the spectroradiometer data. These parameters are static temperature and pressure; longitude, latitude and altitude; pitch, roll and yaw angles; solar zenith and azimuth angles; total ozone columns, and cutoff wavelengths. Except for the last four, these parameters were routinely provided by the aircraft operators. Solar zenith and azimuth angles were calculated based on date, time and aircraft locations. Ozone columns were interpolated temporally and spatially along the flight tracks from assimilated daily global fields of satellite-derived ozone columns (www.temis.nl, Eskes et al. (2003)). Finally, cutoff wavelengths were extracted from the respective lookup-tables (Sect. 2.2.6) using solar zenith angles and total ozone columns as input. Even though the cutoff wavelengths were derived from simulated clear sky downward actinic flux densities, they will be applied in the following under all conditions as well as for upward actinic flux densities that are typically much lower. The data from the lookup tables were taken as safe lower limits for convenience. An unaccounted presence of clouds, for example, would shift cutoff wavelengths towards slightly greater values which is non-critical for the data analysis.

### 3.1.2 Spectral actinic flux densities

During atmospheric measurements several integration times between 3 ms and up to 300 ms were used and the raw data were saved with or without further averaging dependent on the desired time resolution of spectral actinic flux densities. No averaging resulted in a maximum time-resolution of 0.8–0.9 s while 10 s averages typically covered 12–13 single measurements for each integration time. 1 and 3 s averages were finally used for airborne measurements on HALO and the Zeppelin, respectively and 10–60 s averages for ground-based measurements.

Figure 13 shows an example of raw data from a HALO flight at an altitude of 13 km. As was mentioned in the introduction, airborne deployments always comprised simultaneous measurements with two CCD-SR taking separate $2\pi$ measurements of downward and upward actinic flux densities in the upper and lower hemisphere, respectively. In Fig. 13 data from the lower hemisphere are shown on the left hand side (instrument 62000), those from the upper hemisphere are on the right hand side (instrument 62001).

The first step in the data analysis is the subtraction of mean dark signals, usually taken from the corresponding lab calibrations, to remove the dark-signal-induced structures from the spectra (Sect. 2.2.2). For better visibility this is shown in panels (a) and (b) for the UV-B range and the longest integration times of 300 ms. The second step is to apply a linear regression in a wavelength range between 270 nm and the variable cutoff wavelength to determine stray light signals plus any remaining positive or negative offset in dark signals. The dashed lines show the corresponding regression lines. The cutoff wavelength in this example was 291 nm (dotted, vertical lines). The linearly approximated stray light signals were then subtracted resulting in background and stray light corrected signals (blue lines). The lower wavelength limit of 270 nm is not strictly defined and may be adjusted dependent on measurement conditions and the instrument specific shape of the stray light signals, but at least a wavelength range $\approx$10 nm below the cutoff wavelength should be covered in the regression.

Expectedly, downward stray light signals in panel (b) were greater than upward stray light signals in panel (a). On the other hand, as shown in panels (c) and (d) of Fig. 13 the signal fractions caused by stray light are comparable for upward and downward measurements. Moreover, they quickly diminish with increasing wavelengths ($\leq$1% above 320 nm). Accordingly, the uncertainty related with the linear extrapolation of stray light signals to wavelengths of up to 650 nm is insignificant. Tests with various cutoff filters confirmed that there were no significant stray light induced structures in the investigated spectral range. Of course, it cannot be excluded that for other instruments such structures exist or that stray light signals have to be approximated non-linearly. In these cases modified, instrument specific extrapolation procedures have to be developed, as already pointed out in Sect. 2.2.3.

In a third step, spectral calibrations were applied to derive spectral actinic flux densities for each integration time. The corresponding spectra of the examples in Fig. 13 are shown in Fig. 14 in linear and semi-logarithmic representations. In panels (a) and (b) spectra obtained with different integration times were plotted upon each other starting with the shortest integration time. The spectra are virtually congruent except below about 310 nm where deviations for short integration times become apparent because detection limits were approached. Optimized spectra were finally combined by selecting data from the longest available integration time. Any missing data caused by saturation were successively replaced by data obtained with

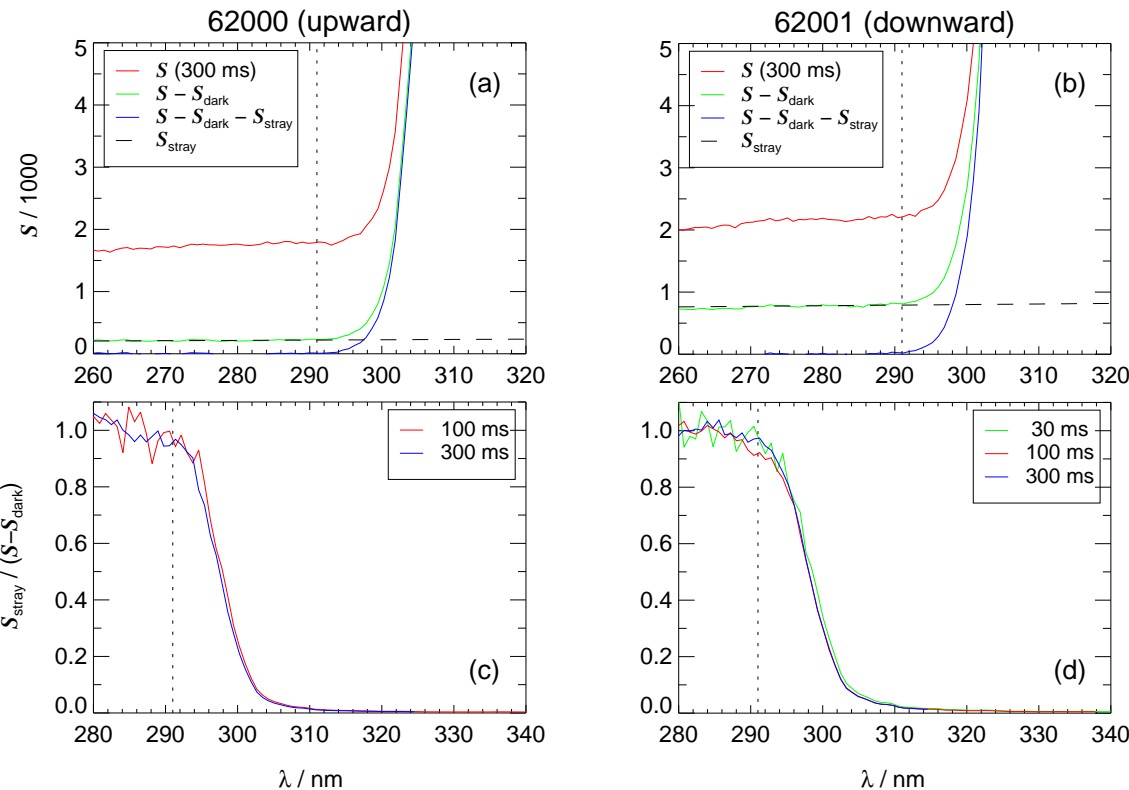

**Figure 13.** Examples of flight raw data and evaluations of instruments 62000 (lower hemisphere, left) and 62001 (upper hemisphere, right). Data were obtained during a HALO flight on 20 Dec 2013 17:30 UTC over the North Atlantic (15.0N, 55.9W, 13.2 km) under conditions with few scattered low-lying clouds (solar zenith angle 47°, ozone column 245 DU). In panels (a) and (b) different colors indicate the evaluation steps: raw data (red), background subtraction (green) and stray light subtraction (blue). Stray light signals (dashed black lines) were determined by linear regression of background corrected signals in a range 270 nm to 291 nm (cutoff wavelength). In panels (c) and (d) the contributions of the inter- and extrapolated stray light signals are shown for the integration times eventually used in the displayed, most effected and atmospherically relevant wavelength range >280 nm.

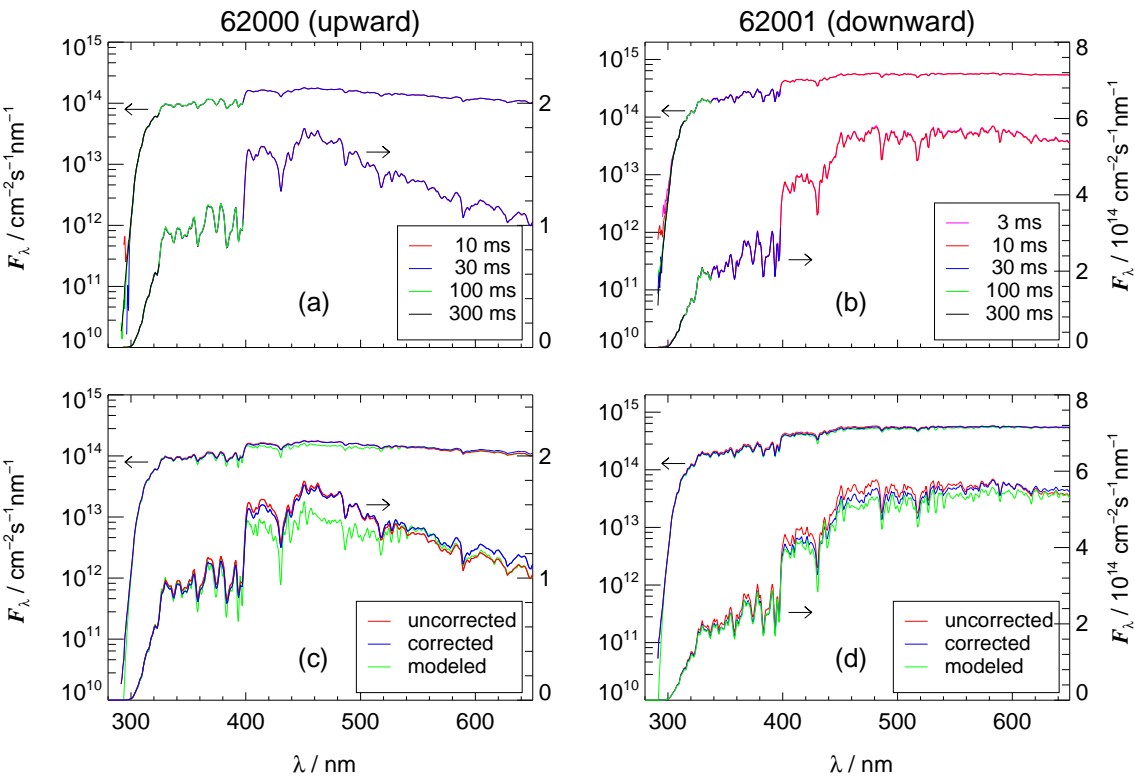

**Figure 14.** Evaluated upward (left) and downward (right) spectral actinic flux densities of the data shown in Fig. 13 in linear and semi-logarithmic representations. Arrows point to the respective axes. In panels (a) and (b) spectra obtained with different integration times are plotted upon each other. Because already for 30 ms (a) and 10 ms (b) no saturation occurred, the data shown for the shortest integration time in each panel were not used for the final optimization of spectra. In panels (c) and (d) the spectra denoted as uncorrected (red) represent the optimized spectra of panels (a) and (b). Optical receiver specific corrections led to the slightly modified, corrected spectra (blue). Results of radiative transfer calculations for clear-sky conditions are denoted as modeled (green).

shorter integration times until the spectrum was complete. Moreover, data below the cutoff wavelengths were set to zero (Sect. 2.2.6). The optimum spectra were then saved for each instrument and this procedure was applied for all measurements along the flight tracks.

In a further step, corrections were made to compensate for imperfections of the optical receivers. These corrections differ
5  for HALO, Zeppelin and ground-based measurements and are usually ranging below 5% with respect to total actinic flux densities, except for airborne measurements close to sunrise or sunset. In addition, data were sorted out where pitch and roll angles exceeded certain limits or where shading of the receivers by aircraft structures may have influenced the measurements. More details on these optical receiver related corrections will be given elsewhere (Lohse and Bohn, 2017). To illustrate the extend of the corrections, uncorrected and corrected spectra are shown in panels (c) and (d) of Fig. 14. Although the corrections
10  are small, wavelength dependencies as well as differences for upward and downward flux densities can be recognized.

As a first assessment, also examples of simulated clear sky actinic flux density spectra are shown in panels (c) and (d). These data were produced with the libRadtran radiative transfer model (Mayer and Kylling, 2005; Emde et al., 2016) taking into account basic flight parameters and the local ozone column. Moreover, a low ground albedo and a low aerosol load were assumed for the measurement location over the North Atlantic. The agreement of measured and modeled downward spectral actinic flux densities is within 5% while there are greater, wavelength dependent differences for the upward component, probably because conditions were not cloud-free underneath the aircraft. A further analysis of these data and comparisons is beyond the scope of the present work. The results merely show that expectedly model and measurements agree for clear-sky downward flux densities.

Total actinic flux density spectra can be produced by adding up data from the upper and the lower hemisphere. However, because the two measurements are independent of each other, this requires a thorough temporal and spectral synchronization, also considering the different instrument response functions. Generally, a separation of upward and downward actinic flux densities is desired, for example for comparison with model calculations as shown in Fig. 14. Total actinic flux density spectra were therefore not produced routinely.

### 3.1.3 Photolysis frequencies

Photolysis frequencies were calculated from actinic flux density spectra according to the examples given in Eqs. 2 and 3 by inserting respective molecular data of the photolysis processes under consideration. Measured spectra and molecular data from the literature were interpolated to a common wavelength grid with 0.1 nm resolution and added up after multiplication (Hofzumahaus et al., 1999; Bohn et al., 2008). The wavelength offsets were also considered in these calculations. Because wavelength offsets were corrected to within 0.05 nm of the true values, the remaining uncertainties are insignificant for the accuracy of photolysis frequencies (Hofzumahaus et al., 1999). The spectral resolutions (FWHM) listed in Tab. 1 are expected to lead to a slight overestimation of $j(O^1D)$ ($\approx 2\%$) and an underestimation of $j(HCHO)$ ($\approx 3\%$) while no significant influence on $j(NO_2)$ is expected according to previous studies (Hofzumahaus et al., 1999; Bohn et al., 2008). $j(O^1D)$ and $j(HCHO)$ are more affected by limitations of spectral resolutions because of the sharp increase of atmospheric actinic flux densities in the UV-B range and narrow spectral features of the HCHO absorption spectrum, respectively.

In accordance with the actinic flux densities, contributions of photolysis frequencies were derived separately for the upper and the lower hemisphere. These contributions were then added up after time synchronization to obtain the photochemically relevant total photolysis frequencies.

### 3.2 Research flight example

An example of photolysis frequencies obtained during a research flight with HALO is shown in Fig. 15 together with some flight related data. The route led from Oberpfaffenhofen in Germany (48.08N, 11.28E) to the island of Barbados (13.07N, 59.49W) in the Western Atlantic Ocean and was part of the NARVAL campaign conducted in December 2013 and January 2014 (Klepp et al., 2014). A map of the flight route can be found in Fig.16. Flight altitudes were 12–14 km during the main part of the transfer with the aircraft presumably above any underlying clouds. The high altitude also resulted in low ambient

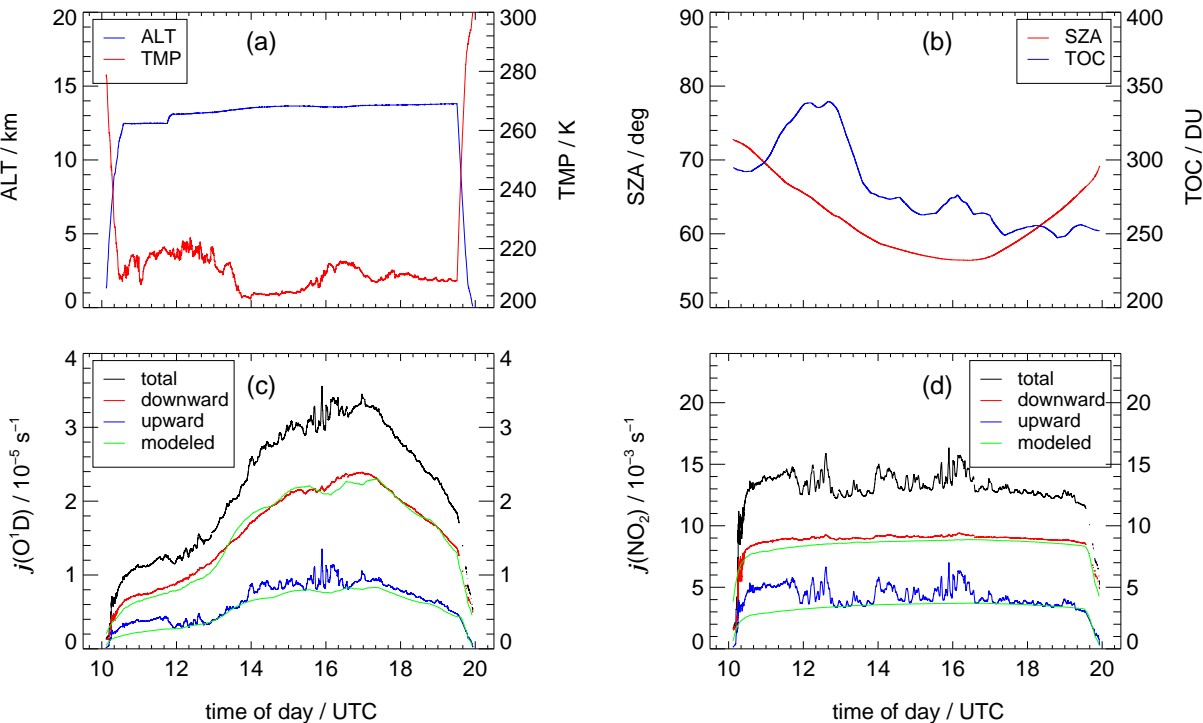

**Figure 15.** Example data from a HALO research flight on 19 Dec 2013 from Oberpfaffenhofen (Germany) to Barbados. Altitude (ALT) and static temperature (TMP) in panel (a), as well as solar zenith angles (SZA) and total ozone columns (TOC) in panel (b), are important boundary conditions and parameters used in the data evaluation. Photolysis frequencies $j(O^1D)$ and $j(NO_2)$ in panels (c) and (d) were calculated for the corresponding static temperatures. Rapid fluctuations of upward components of photolysis frequencies were induced by underlying clouds. Modeled data are clear-sky upward and downward components of photolysis frequencies from radiative transfer calculations.

temperatures around 210 K. Owing to the season and the times of day, solar zenith angles were changing in a narrow range of 55–75° even though the destination was at much lower latitudes. Also shown in Fig. 15 are the interpolated total ozone columns along the flight track from satellite data that exhibit a typical decrease of ozone columns towards lower latitudes.

For the photolysis frequencies $j(O^1D)$ and $j(NO_2)$ the directly measured downward and upward components are shown in

5   Fig. 15 as well as the photochemically relevant total values. Downward photolysis frequencies exhibit a smooth diurnal variation typical for cloud-free conditions above the aircraft. In contrast, upward photolysis frequencies show stronger, sometimes rapid fluctuations caused by underlying clouds. The contributions of upward radiation are comparable for $j(O^1D)$ and $j(NO_2)$ but differ in detail. Moreover, diurnal variations are strongly different for the two photolysis frequencies because of a more distinct dependency of $j(O^1D)$ on solar zenith angles. In addition $j(O^1D)$ was strongly influenced by total ozone columns

10   which can be recognized in the first part of the flight.

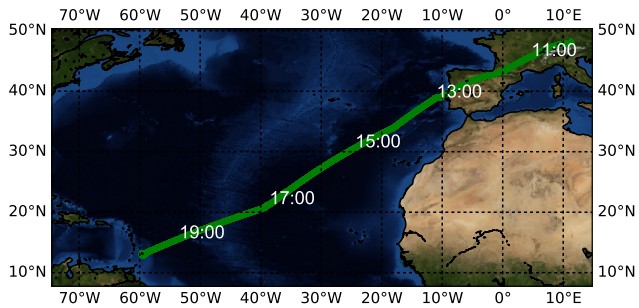

**Figure 16.** Map of the flight route of HALO on 19 Dec 2013 from Oberpfaffenhofen, Germany to the island of Barbados. Indicated times are UTC.

The effect of ambient temperature is also more pronounced for $j(O^1D)$. Compared to a reference temperature of 298 K, in this example, $j(O^1D)$ and $j(NO_2)$ are on average smaller by factors of 0.73 and 0.92, respectively. These numbers are based on the temperature dependence of the molecular data from the literature (Daumont et al., 1992; Matsumi et al., 2002; Merienne et al., 1995; Troe, 2000) and demonstrate the importance of ambient temperature for aircraft measurements.

5    Photolysis frequencies calculated from libRadtran simulated actinic flux density spectra along the flight track are shown for comparison in panels (c) and (d) of Fig. 15. The agreement is satisfactory but not perfect during all parts of the flight. In particular periods with underlying clouds can be recognized when the clear-sky model underestimates the upward component. A further analysis of spectral actinic flux densities or photolysis frequencies obtained during airborne missions is outside the focus of this work and will be given elsewhere.

## 10    3.3    Evaluation by ground-based comparisons

In order to evaluate the accuracy of photolysis frequencies obtained in field measurements, ground-based comparisons of the CCD-SR with a double-monochromator based reference instrument (DM-SR) were routinely made. Typically before and after a deployment, the instruments were set up on a roof platform for parallel measurements of downward spectral actinic flux densities for a couple of days.

15    The reference instrument was described in detail elsewhere (Hofzumahaus et al., 1999; Bohn et al., 2008). As was mentioned in the introduction, the main advantage of the DM-SR is an effective stray light suppression. However, the scanning procedure leads to a limited time resolution and reduced accuracy under variable atmospheric conditions. To minimize this limitation, the DM-SR measurements were confined to a wavelength range 280–420 nm, the spectral range most important for the determination of photolysis frequencies. This resulted in a time-resolution of about 2 min. The instrument was operated at
20    a FWHM of 1 nm, i.e. the spectral resolution was slightly better than that of the CCD-SR (Tab. 1).

### 3.3.1 Comparison of spectral actinic flux densities

Figure 17 shows examples of actinic flux density spectra obtained simultaneously with the DM-SR and instrument 62001. The spectra were selected for stable, clear-sky conditions to avoid deviations caused by DM-SR scanning operations. The CCD-SR spectrum is a 10 s average obtained with a maximum 300 ms integration time. Minor optical receiver specific corrections ($\approx$2% in this case) were already included for both instruments. Panel (a) shows the expected sharp increase of actinic flux densities in the UV-B range that is reproduced similarly by both instruments. Also shown is a radiative transfer model spectrum from the set of spectra produced to derive the cutoff wavelengths for ground measurements. The spectrum was selected for the indicated ozone column and solar zenith angle. In panel (b) the comparison is extended to the complete spectral range covered by the DM-SR. Generally good agreement is obtained except for sharp spectral features that are resolved more accurately by the reference instrument because of a smaller FWHM. The agreement of the modeled spectrum with the DM-SR is better because in the model calculations a matching FWHM of 1 nm was used. Error bars in panels (a) and (b) correspond to total uncertainties of 5–6% for both DM-SR and CCD-SR based spectral calibration uncertainties plus a 2% uncertainty from the optical receiver corrections. Additional uncertainties from instrument noise and stray light effects are invisible in panels (a) and (b).

Panel (c) of Fig. 17 shows the increase of UV-B spectral actinic flux densities in a semi-logarithmic plot where more details can be recognized. Here also data below the cutoff wavelength (vertical line) that are usually set to zero for the CCD-SR are shown for comparison. In this wavelength range data scatter around zero as expected, albeit with different residual noise. For the DM-SR the noise is unaffected by stray light and similar to nighttime values corresponding to a noise equivalent spectral actinic flux density $F_\lambda^{\mathrm{NE}} \approx 1 \times 10^9$ cm$^{-2}$ s$^{-1}$ nm$^{-1}$. Thus the DM-SR noise is smaller by a factor of about 10 compared to the $F_\lambda^{\mathrm{NE}}$ of the CCD-SR in the 280–290 nm range (Fig. 11) and the residual noise of the CCD-SR is accordingly greater. From the measured signals, $F_\lambda^{\mathrm{NE}} \approx 1 \times 10^{10}$ cm$^{-2}$ s$^{-1}$ nm$^{-1}$ were derived for the CCD-SR using Eq. 6 and taking into account 10 s averaging. These values were added to or subtracted from the data in Fig. 17. Instead of error bars which cannot be reproduced in this representation, envelopes are shown comprising color coded positive and negative values for both instruments. A comparison with the radiative transfer modeled data shows that these are reproduced to well below the cutoff wavelength by the DM-SR while for the CCD-SR the detection limit is reached slightly above the cutoff wavelength. It should be noted that the $\approx$6% uncertainties shown in panels (a) and (b) are invisible in panel (c).

Considering that 10 s averages are shown in Fig. 17 the $F_\lambda^{\mathrm{NE}}$ are in fact greater by a factor of about two compared to the data shown in Fig. 11. The reason is that for this particular deployment the CCD-SR sensitivity was reduced by a factor of $\approx$0.6 because of an increased fiber length compared to the data shown in Fig. 8. Moreover, even though stray light signals were subtracted, they induced some shot noise in addition to the dark noise. This increased the total noise by about 25% under local noon and clear-sky conditions which is the maximum increase expected by shot noise from stray light under all conditions.

The subtraction of stray light signals ($\approx$600 around noon) led to no appreciable increase of the noise even though these signals correspond to actinic flux densities of $1.5 \times 10^{12}$ cm$^{-2}$ s$^{-1}$ nm$^{-1}$ around 300 nm. It can be concluded that the interpolated stray light signals below the cutoff wavelength are well within 1% of the true values. If a corresponding additional uncertainty is assumed for all wavelengths, total $F_\lambda$ uncertainties can be derived that range between $\Delta F_\lambda \approx 0.06 \times F_\lambda + F_\lambda^{\mathrm{NE}} + 2 \times 10^{10}$ cm$^{-2}$

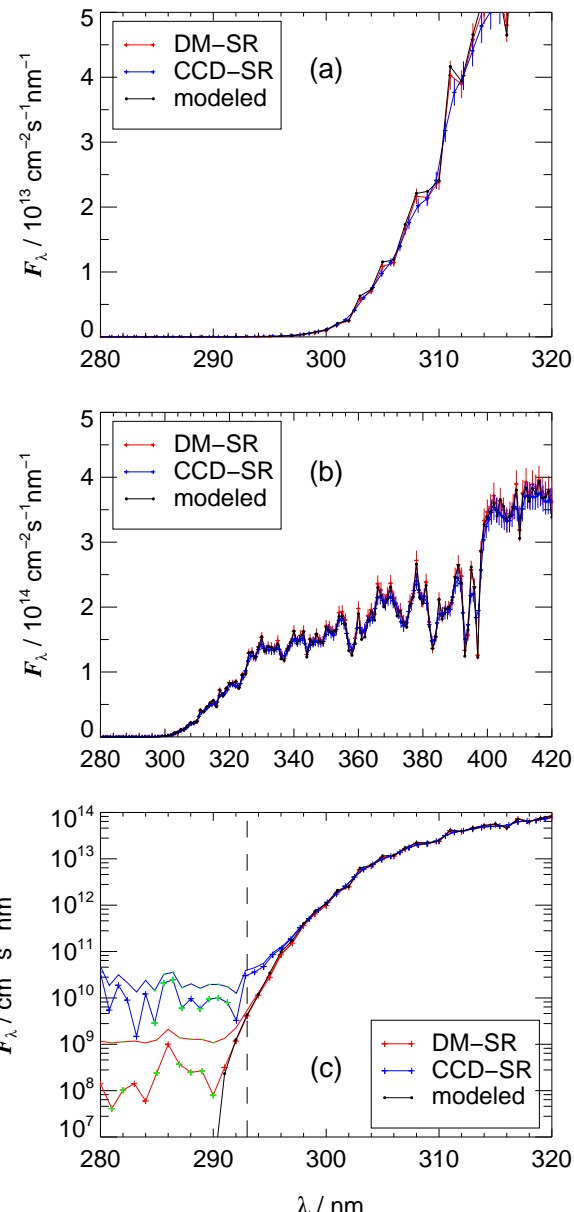

**Figure 17.** Comparison of actinic flux density spectra obtained on the ground with a double-monochromator based reference instrument (DM-SR) (red) and instrument 62001 (blue). Measurements were made on 01 Aug 2013 at Jülich (Germany) under clear-sky conditions. The spectra were taken around 12:00 UTC at a solar zenith angle of 33° and an ozone column of 310 DU. Black data points show the results of radiative transfer calculations for the same conditions. The different representations emphasize the increase of actinic flux densities in the UV-B range in panel (a) and the dynamic range of data in panel (b). Green data points in the semi-logarithmic plots of panel (c) show negative values that were plotted at their absolute values to make them visible. Full lines with minimum symbol size indicate the corresponding upper or lower limits after addition or subtraction of instrument noise. The dashed vertical line shows the cutoff wavelength below which values of the CCD-SR were normally set to zero. Note the different spectral actinic flux density and wavelength ranges.

$s^{-1}$ $nm^{-1}$ around noon and $\Delta F_\lambda \approx F_\lambda^{NE}$ after sunset when stray light and atmospheric signals vanish. Examples of $F_\lambda^{NE}$ for typical aircraft measurements are given in Tab. 4. The $F_\lambda^{NE}$ generally depend on wavelength, atmospheric conditions, spectral sensitivities, integration times and averaging (if applicable) (Sect. 2.2.7). The maximum stray light contribution of $2\times10^{10}$ $cm^{-2}$ $s^{-1}$ $nm^{-1}$ to total uncertainties is comparable to that of the CCD dark noise which only plays a role for very

low $F_\lambda \leq 10^{12}$ $cm^{-2}$ $s^{-1}$ $nm^{-1}$. The 1% assumption obviously is a rough estimate which may have to be adjusted for other instruments. In any case, the stray light signals approximately follow a cosine dependence on SZA and their relative importance increases with increasing SZA, at least for short wavelengths around 300 nm. Moreover, stray light signals are significantly greater in the presence of direct sunlight, i.e. the uncertainty would be lower for example for the upward $F_\lambda$ shown in Fig. 14 or in the presence of clouds. Generally, the stray light influence on $F_\lambda$ uncertainties is difficult to assess because it depends on

instrument properties, measurement conditions and the procedure how stray light signals are determined.

### 3.3.2   Comparison of photolysis frequencies

The overall performance of the CCD-SR was evaluated by a comparison of photolysis frequencies during the comparison periods. Figure 18 shows an example of correlation plots and ratios of photolysis frequencies as a function of solar zenith angles and the reference values for the most critical $j(O^1D)$. The scatter visible in the correlation plots is caused by the

DM-SR scanning operation under variable atmospheric conditions, i.e. by the presence of moving clouds, in particular under broken-cloud conditions. These variations can go in both directions and cancel each other out over longer periods, i.e. they do not influence the slope of linear regressions. This is demonstrated in the Supplement where a virtually scatter-free subset of the data selected for clear sky conditions is shown for comparison. However, because clear-sky conditions are rare at Jülich, Fig. 18 shows a typical comparison result. The slopes of the regressions are taken as a measure for the agreement of the

measurements. Table 6 gives an overview of regression line slopes from all ground-based comparisons associated with various airborne deployments for $j(O^1D)$ and $j(NO_2)$. Including optical receiver corrections, deviations from unity are typically within $\pm5\%$. The remaining discrepancies are attributed to uncertainties of the laboratory and in-field calibrations, optical receiver corrections and differences in spectral resolutions. The agreement is well within the estimated 5–6% of combined uncertainties of spectral calibrations and receiver corrections that apply to both types of instruments.

No linearity problems are evident for CCD-SR or DM-SR measurements but the plots in Fig. 18 of $j(O^1D)$ ratios as a function of solar zenith angles and $j(O^1D)$ reveal increased scatter towards low sun or low $j(O^1D)$ when the detection limits of the instruments were approached. For comparison CCD-SR data are shown where spectral actinic flux densities below the cutoff wavelengths were set to zero (blue) and where this was not made (red). In the latter case scatter is apparently greater because of higher CCD-SR detection limits in accordance with the results of Sect. 2.2.6. In fact, for the DM-SR nighttime $j(O^1D)$ standard

deviations of $8\times10^{-9}s^{-1}$ were obtained while for instrument 62001 $2\times10^{-9}s^{-1}$ and $4\times10^{-8}s^{-1}$ resulted with and without setting flux densities to zero below cutoff wavelengths, respectively. The data for the CCD-SR are in reasonable agreement with the predictions in Tab. 5 considering 10 s averaging and the lower sensitivity mentioned above. The scatter of the red data points in Fig. 18 is therefore mainly caused by the detection limit of the CCD-SR while that of the blue points is dominated by the detection limit of the DM-SR. In any case, disregarding data below the cutoff wavelengths has little effect for $j(O^1D)$

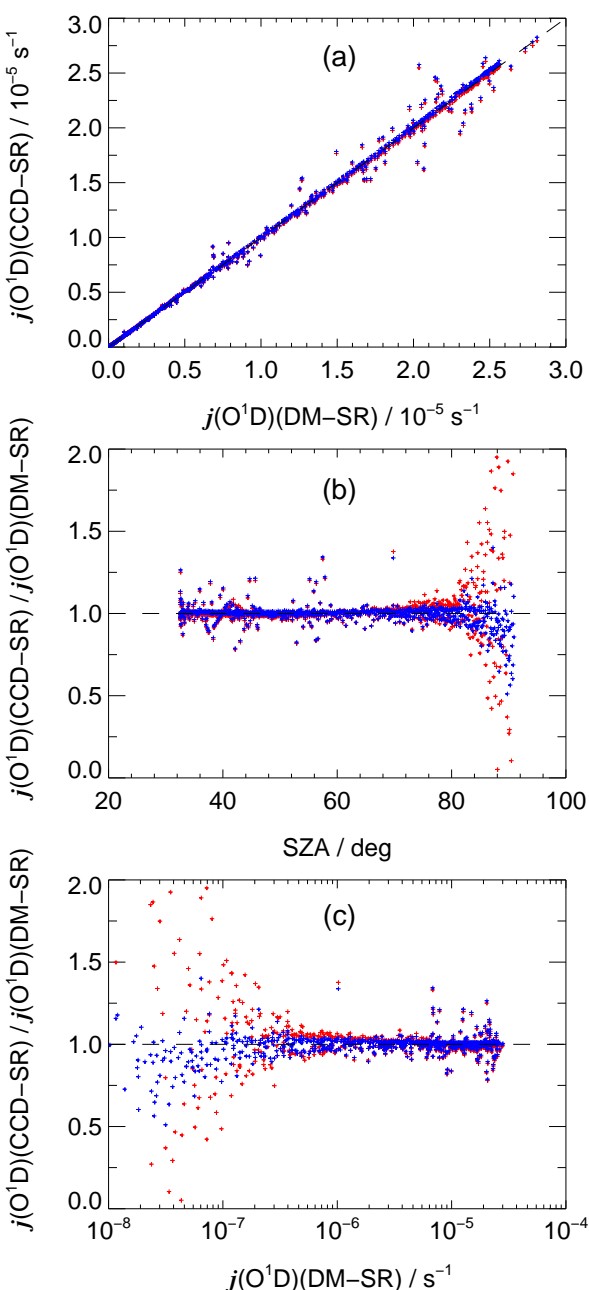

**Figure 18.** Comparison of $j(O^1D)$ photolysis frequencies obtained on the ground with a double-monochromator based reference instrument (DM-SR) and instrument and 62001. Measurements were made during the period 30 Jul – 01 Aug 2013 at Jülich (Germany). Panel (a): correlation plot. Panel (b): ratios as a function of solar zenith angles. Panel (c): ratios as a function of reference values (DM-SR). Dashed lines indicate 1:1 relationships. Scatter visible in the correlation plot (upper panel) is caused by clouds, i.e. by synchronisation issues. Additional scatter in the ratios towards large SZA (middle panels) and low $j(O^1D)$ (lower panels) is caused by different detection limits. Blue and red data points were obtained when spectral actinic flux densities below cutoff wavelengths were set to zero and not set to zero, respectively.

greater than about $5\times10^{-7}\mathrm{s}^{-1}$. Therefore this procedure is recommended but not crucial unless very small values of $j(\mathrm{O^1D})$ are of interest. The nighttime standard deviations mentioned above correspond to $j(\mathrm{O^1D})$ detection limits of about $6\times10^{-9}\mathrm{s}^{-1}$ and $1.2\times10^{-7}\mathrm{s}^{-1}$, corresponding to no more than around 0.015% and 0.3% of typical summer noontime values, respectively.

For $j(\mathrm{NO_2})$ nighttime standard deviations of $2.4\times10^{-8}\mathrm{s}^{-1}$ and $1.6\times10^{-7}\mathrm{s}^{-1}$ were obtained for the DM-SR and the CCD-SR 62001. For the CCD-SR the value is greater than predicted by considering dark noise alone (Tab. 5). This is attributed to the fact that the determination of residual dark signals was optimized for the UV-B range and that for greater wavelengths these residuals were extrapolated. Improvements on the $j(\mathrm{NO_2})$ detection limits of the CCD-SR could probably be obtained by expanding the concept of cutoff wavelengths above 310 nm and for SZA>90°. On the other hand, there seems to be no need for such an improvement because values below the current detection limit of about $5\times10^{-7}\mathrm{s}^{-1}$ are considered insignificant.

In order to assess the additional uncertainties of photolysis frequencies produced by stray light signal subtractions, $j(\mathrm{O^1D})$ and $j(\mathrm{NO_2})$ were calculated from pseudo actinic flux density spectra derived from subtracted stray light signals on a clear-sky day. This resulted in maximum stray light affected $j(\mathrm{O^1D})$ of around $1\times10^{-5}\mathrm{s}^{-1}$ and $6\times10^{-5}\mathrm{s}^{-1}$ with and without application of cutoff wavelengths, and a maximum stray light induced $j(\mathrm{NO_2})$ of around $4\times10^{-5}\mathrm{s}^{-1}$. Assuming a 1% uncertainty for the subtractions, total uncertainties of photolysis frequencies can be estimated. Taking into account the noise induced photolysis frequencies of the aircraft measurements in Tab. 5, total $j(\mathrm{O^1D})$ uncertainties with cutoff wavelengths range between $\Delta j(\mathrm{O^1D}) \approx 0.06 \times j(\mathrm{O^1D}) + 1.3\times10^{-7}\ \mathrm{s}^{-1}$ around noon and $\Delta j(\mathrm{O^1D}) \approx 4\times10^{-9}\ \mathrm{s}^{-1}$ in the dark. The corresponding numbers without cutoff wavelengths are $\Delta j(\mathrm{O^1D}) \approx 0.06 \times j(\mathrm{O^1D}) + 7\times10^{-7}\ \mathrm{s}^{-1}$ around noon and $\Delta j(\mathrm{O^1D}) \approx 1\times10^{-7}\ \mathrm{s}^{-1}$ in the dark. The variable noise equivalent photolysis frequencies generally depend on atmospheric conditions, spectral sensitivities, integration times and averaging (if applicable) as explained in Sect. 2.2.7. For the stray light induced contributions it should be noted that despite greater absolute values at small SZA, the relative importance of the uncertainties increase with SZA, reaching maxima of 3% and 100% of $j(\mathrm{O^1D})$ around SZA=87° with and without cutoff wavelengths. Thus, the use of cutoff wavelengths also helps to confine the stray light influence on $j(\mathrm{O^1D})$. For $j(\mathrm{NO_2})$, noise and stray light induced uncertainties are negligible before sunset and the stray light induced fraction vanishes after sunset.

### 3.3.3 The stray light issue

Apparently, the higher level of stray light of the CCD-SR compared to the DM-SR is no major obstacle to derive accurate photolysis frequencies, including $j(\mathrm{O^1D})$, at least for the applied type of instrument. Jäkel et al. (2007) came to the same conclusion but their preferred method of stray light correction was slightly different for the UG5-filtered instrument. It was based on a stray light spectrum obtained with a calibration lamp and a 700 nm cutoff filter that for each spectrum was scaled for matching averages in the 270–290 nm range. However, for instruments without UG5 filter this procedure is no option because the spectral shape of the stray light is strongly influenced by radiation blocked out by a 700 nm filter. Moreover, the slope of the stray light signals during lamp calibrations was usually slightly negative while in the atmosphere they were typically slightly positive. For that reason laboratory measurements were not consulted to estimate the shape of atmospheric stray light signals also because the blackbody temperature of the calibration lamps is much lower than that of the sun. This may explain why the stray light contributions in Fig. 13 diminish much faster than those in Fig. 8 and corresponding figures in the Supplement.

**Table 6.** Results of ground-based spectroradiometer comparisons associated with contemporary airborne instrument deployments of different CCD-SR. The numbers are slopes of $j(\mathrm{O^1D})$ and $j(\mathrm{NO_2})$ regression lines based on 2-5 day parallel measurements with a DM-SR reference.

| deployment | instruments | slope (instrument vs. reference) | |
|---|---|---|---|
| | top / bottom | $j(\mathrm{O^1D})$ | $j(\mathrm{NO_2})$ |
| HALO 2010 | 62001 / 62000 | 1.003 / 0.997 | 0.956 / 0.972 |
| Zeppelin 2012 | 45853 / 62001 | 1.034 / 1.054 | 0.998 / 0.979 |
| Zeppelin 2013 | 62008 / 85235 | 1.001 / 1.035 | 0.978 / 0.981 |
| HALO 2013 | 62001 / 62000 | 1.005 / 0.969 | 0.964 / 0.955 |
| HALO 2015 | 62001 / 62000 | 1.019 / 0.959 | 0.981 / 0.957 |

Fitting the spectral dependence of measured stray light signals in a condition dependent range defined by the atmospheric cutoff wavelengths was therefore a manifest approach. Because the results were satisfactory no alternative procedures were systematically tested. The practice is believed to be widely transferable to spectral irradiance measurements.

More sophisticated stray light correction methods were developed in the past for array spectroradiometers using tunable light sources to investigate the instrument's stray light response as a function of wavelength. With this instrument-specific information, a correction can be made based on the measured spectra alone (Zong et al., 2006). However, for the instruments described in this work, a substantial fraction of the stray light comes from a spectral region beyond the measured range and this unaccounted fraction is strongly variable dependent on the presence or absence of direct sunlight. Because the same problem arises for atmospheric spectral irradiance measurements in the UV-B range, the method by Zong et al. (2006) was refined for these applications (Kreuter and Blumthaler, 2009; Nevas et al., 2014). However, a recent blind inter-comparison of spectral UV irradiance and UV-index measurements revealed that these methods are not consistently adhered to and accurate stray light corrections remain a complicated and critical issue (Egli et al., 2016). In this work we showed that a comparatively simple approach led to satisfactory results for the investigated type of instruments.

## 4 Conclusions

Spectral actinic flux densities can be measured with high accuracy and high time resolution in the atmospherically relevant UV/VIS range using CCD array spectroradiometers. Because the instruments are compact and mechanically robust, they are suitable for high quality airborne measurements. In this work, we investigated the key properties of a widely used instrument type in the laboratory, developed a straightforward method for calibrations with irradiance standards, and derived a scheme to evaluate field measurements under variable atmospheric conditions. The major difficulties were accurate measurements in the UV-B range because calibrations and field measurements are affected by the notorious stray light problem which is typical for single-monochromator applications. We showed that this problem can be widely resolved in the laboratory by the use of long path cutoff filters including additional corrections, and during field measurements by utilizing the variable natural long

path cutoff provided by the stratospheric ozone layer. Ground-based field comparisons with a double-monochromator reference instrument confirmed the practicality of the approach for atmospheric measurements. Even though the stray light effects do not completely vanish they can be contained so that they become insignificant for the determination of photolysis frequencies, including $j(O^1D)$. However, it should be noted that the results of this work refer to the radiometric part of the determination of photolysis frequencies. Additional uncertainties exist that are related with molecular parameters of photolyzed species. These uncertainties are process specific and are substantial for many photolysis processes mainly because quantum yields are poorly known. Because of extreme temperature conditions this problem should be kept in mind, in particular for airborne applications. Optical receiver issues are also more pronounced for airborne measurements but can be dealt with independently because they are not directly related with the type and performance of the spectroradiometer employed.

*Acknowledgements.* The authors thank a great number of people who helped to get instruments airborne on the platforms HALO and Zeppelin NT. We thank Bernhard Mayer, Arve Kylling and co-workers for making the libRadtran radiation transfer model available to the scientific community. Public provision of total ozone column data by the TEMIS/ESA team is gratefully acknowledged. We thank the Sensor and Data Group of DLR Flight Experiments department for providing HALO related data and Lisa Beumer (FZJ) for preparing the map of the HALO example flight. Finally, we thank the Deutsche Forschungsgemeinschaft for funding under grant BO 1580/4-1.

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
