# Peer review of "Calibration and evaluation of CCD spectroradiometers for ground-based and airborne measurements of spectral actinic flux densities"

_Atmospheric Measurement Techniques, 2017_

## Referee Comment (RC1) · Anonymous Referee #1 · 4 Apr 2017

Calibration and evaluation of CCD spectroradiometers for airborne measurements of spectral actinic flux densities

This manuscript describes the calibration procedures and data evaluation of a CCD spectroradiometer (CCD-SR). The instrumental characteristics are partly compared with results from other publications. The performance of the CCD-SR is demonstrated for airborne and ground-based observations.

The manuscript is very technical and gives a lot of details. It is well-written but could be shortened at some points, since it is no new instrumental techniques described here. The benefits for the reader should be pointed out more clearly in the introduction. What is new compared to former publications? The manuscript is recommended for

publication in AMT. Nevertheless, the following comments should be addressed first.

General comments:

1. The CCD-SR shows good performance also for ground-based observations. The disadvantage of less sensitivity and impact of stray light in the UV spectral range can be avoided by careful calibrations and some corrections as written in the paper. The comparison with the DM-SR has shown sufficient agreement. This precise scanning instrument, however, has much lower time resolution than the CCD-SR which is not sufficient for fast changing atmospheric conditions (as cloudy sky). For this reason I suggest to change the title: "Calibration and evaluation of CCD spectroradiometers for ground-based and airborne measurements of spectral actinic flux densities". There is no need to restrict the application of the CCD-SR to airborne measurements. That could be made clear by extending the ground-based comparison (see comment #41).

2. Since the angular response of the entrance optics is highly affecting the quality of the data it would be worthwhile to implement it in this paper. In this case some changes to the text and text cuts should be made not to overload the manuscript when adding the discussion about the angular response. That would give the reader a full description of the instrumental performance in one paper.

Specific comments:

3. Give wavelength range of the CCD-SR in the abstract.

4. p1l7: "flux densities in a 300 nm range" – What does it mean? Is it the detection limit at 300 nm wavelength?

5. p1l17: "below cutoff wavelength" – Not knowing the manuscript, the reader will not know what "cutoff wavelength" means. Better write something like that "below atmospheric cutoff wavelength which was simulated depending on solar zenith angle, ozone column, . . ."

6. p1l17: "with reference instrument data" – Which instrument? Tell the reader that is

a high sensitive double monochromator.

7. p1l19: "Overall, the investigated instruments are clearly ..." – already mentioned in the beginning of the abstract (l3).

8. p2: Please discuss the dependence on temperature and pressure of molecular parameters.

9. p2l1: Explain shortly what is meant by "stray light". Furthermore, mention the lower sensitivity in the UV-B.

10. p3l3: "upward radiation in the UV range can often be neglected" – only valid over dark surfaces

11. p3l14: "have low time resolution" – What means low? Please give numbers.

12. p5l6: "The idea is to piece together ..." – I am wondering if there is mismatch between the spectra sampled over different integration times, because the measurement volume is different. In particular for fast changing atmospheric conditions in combination with the speed of HALO, I would expect some differences. Did the authors compare the spectra for some examples without oversaturation? Furthermore, the linearity of the detectors should be shortly discussed.

13. p5l8: ".. calculation of final actinic flux ..." – In fact, these data are only a kind of quicklook. Post-processing is required as mentioned in the subsequent sentence.

14. p5l12: Maybe a flow chart of the following calibrations would be helpful.

15. p5l22: "dark spectra" – Explain shortly.

16. p5l23: "were fitted with an " – Why did the authors did not use also the Neon emission lamp which covers the longer wavelengths greater than 546 nm?

17. p9l4: "Noise levels were found to increase with the square root of signals ..." - Can you show a plot for illustration, maybe for different wavelengths?

18. p10l1: "example of signal-to-noise ratios" – Give equation. I am wondering if it is correct to illustrate the wavelength dependence of the SNR this way. Since the SNR depends on the magnitude of the signal itself, a constant signal should be used for all wavelengths (however feasible) to demonstrate the pure wavelength dependence. As suggested in comment #17, plot the SNR as a function of the signal for different wavelengths.

19. p10l8: "Laboratory calibration" – include radiometric

20. p10l16 "shows little wavelength dependence" – What means little, give numbers.

21. p11 Fig. 5: What is the gain of information showing the procedure for two integration times? The reader might be more interested in the stray light fraction than showing each step of the data evaluation.

22. p12l12: "WG320, WG360" – Please refer to Schott, otherwise it is not clear where the "WG" comes from.

23. p12l23: "... stray light quickly diminishes with increasing wavelength..." As mentioned in comment #21, this would be good to see in a plot for all spectroradiometers.

24. p13 Fig. 6: Here, a test of the linearity could be implemented.

25. p13l14: It is suggested to delete the paragraph. After introducing the spectral calibration, the reader assumes that the wavelength shift is considered in the radiometric calibration.

26. p14l11: Also here, delete the paragraph. The reader will not be interested in the Metcon software.

27. p16l14: "in total uncertainties between ..." - The error due to angular response is not included here as it should be when talking about the total uncertainty.

28. p17l12: "Detection limits are usually defined as the three-fold ..." – Please give a reference.

29. p17l20: "the noise limits correspond to 0.1% and 0.0001%..." – Are these values derived from the table? Give the reference (typical maximum) values. Tab. 3 lists also noise equivalent photolysis frequency for a limited spectral range above 300 nm in brackets. A better motivation might be to give the results for two situations (low SZA, low total ozone column vs. high SZA, high total ozone column). Note the conditions in the table caption. It would give the reader a smooth introduction to the cutoff wavelength paragraph.

30. p17l24-p17l31: This paragraph could be deleted. It was already stated in the introduction that in the UV-B the ozone photolysis is more affected than the NO2-photolysis.

31. p17l32: "cutoff-wavelength" - atmospheric cutoff wavelength. The authors created some look-up-tables (as mentioned on p18l3). A contour plot showing the dependence of the cutoff wavelength on SZA and total ozone column would be nice to see.

32. p19l14: How are temperature and pressure variations considered in the look-up-tables? There should be some sensitivity in particular for ozone photolysis.

33. p19l21-p21l23: This section can be shortened. A flow chart summarizing the steps to create a spectrum from raw data might be helpful.

34. p19l25 Is the dark signal taken from laboratory measurements? If yes, how are changes be considered during the campaign (p8l1)?

35. p20 Fig. 10: The upper two panels are sufficient to illustrate the stray light and offset correction for field measurements. An additional subfigure showing the stray light fraction would be more interesting than the remaining six panels (supporting also p22l2).

36. p21 Fig. 11: Combine with Fig.10., the logarithmic representation would be sufficient. Could the authors add a comparison with simulations for the downward actinic flux density?

37. p22l8: "starting with the longest available..." – As mentioned earlier the reader

might be more convinced of this combination method by showing spectra measured for different integration times in one plot with indication at which wavelengths the final spectrum were merged.

38. p23l3: The reader probably expects the results of the photolysis frequency right here.

39. p23l16: One example is sufficient, either 62000 or 62001. When comparing with DM-RS add the measurement uncertainty for both instruments in Fig. 12.

40. p24 Fig.12: The lower left panel would be sufficient. Could the authors here also some simulated spectrum?

41. p25l1: The authors compare j-values derived from CCD-SR and DM-SR for two days including cloud periods. Here the advantage of the better time resolution could be directly shown by a time series for of j-values with best temporal resolution for each instrument. Furthermore, filtering these cloud events from Fig. 13 might be worthwhile to exclude the data sampled during variable conditions within the time frame of one DM-SR measurement.

42. p29 Fig. 14: Show a map instead of the time series of latitude and longitude. Combine SZA and total ozone column in one panel. Remove date in x-axis in the lower panels. Add some simulated j-values for downward components.

Technical comments:

43. p2l13, p2l22: "spectral actinic photon flux density" – delete "photon"

44. p2l26: "easyness" - easiness

45. p3l1: "stray-light" - stray light

46. p3l16: "maneuvers in flight" - flight maneuvers

47. p4l7: "properties" - characteristics

48. p4l14: "lengths" - length

49. p5l6: "The idea is to piece together ..." - combine

50. p5l12: "characterisation" - characterization

51. p6 Fig. 2: Rearrange plot to save some space. Maybe one combined color coded plot is sufficient showing the spectra as function of wavelength difference referred to center wavelength (+/- 4 nm). Otherwise, please give labels for each subfigure (a) –(e).

52. p8 Fig. 3: Unit of signal? Counts?

53. p8l13: "is considered a combination" - "is considered as a combination"

54. p9 Tab. 2: Numbers in Counts?

55. p9 Fig. 4: Label subfigures and use larger points. Plot subfigures in two columns.

56. p12l8 and hereafter: Maybe better to use the index "stray" instead of "scat" in the S-variable.

57. p16 Fig. 8: Figure caption "hangar measurements" - field measurements, label the subfigures with (a) - (d)

58. p16l2: "Firstly by the ..." – no sentence, link it to the sentence before by using ";"

59. p17l14: "be considered a theoretical" - as a theoretical

60. p17l18: "are listed in Tab. 2" - Tab. 3

61. p17l25: "Tab. 2" - Tab. 3

---

## Referee Comment (RC2) · Anonymous Referee #2 · 2 May 2017

Summary:

This paper characterizes the CCD-SR instrument that deploys on the HALO aircraft and continues the discussion of actinic flux measurements and calibrations using Metcon CCD-based spectrometers. The determination of spectral and wavelength assignment accuracies and sensitivities are discussed and a new technique for removing stray light was demonstrated to improve UV-B spectral accuracy and account for low limits of detection. Cutoff wavelengths are determined for each measured spectra based on radiative transfer modeled fluxes to reduce noise in photolysis products and examine stray light impacts. Ground-based comparisons with low-stray light instrumentation show strong agreement to near the detection limits. The resulting data are shown

to be of sufficiently high accuracy for studies of actinic flux and calculated photolysis frequencies. This work is of value, well written and recommended for publishing after addressing the comments.

General comments:

P3 L3-4: The authors contend that "upward radiation in the UV range can often be neglected" at surface sites. Upwelling jNO2 is often 4-10% to the total photolysis and can be much greater. The upward UV radiation can rarely be neglected (though it can be estimated from the downwelling). This sentence should be dropped or adjusted.

P12, L24-27: The authors may have some luck in that they found linear stray light structure in the UV-B in 4 of the 5 tested spectrometers (with the exception of 45853). The paper's entire stray light analysis relies on a linear-based subtraction. However, non-linear stray light response is not fully controlled for in the manufacturing and is determined by testing after production. Thus, I think they should state explicitly that determination of the stray light spectral structure is required to assess if a linear regression is appropriate.

P19, L13-15: The lookup tables are appropriate for clear-sky determinations of cutoff wavelengths, but no mention is made of the impacts of clouds and aerosols. Extinction by clouds and aerosols can greatly reduce the in situ measured flux and cutoff wavelength would be higher than expected from the table. In these cases, the variability in cutoff could be larger than 2 nm (as mentioned on P18 L5-6). Ideally the measured spectra could be evaluated for the cutoff, in concert with the model. The measurements are often most interesting in these complex atmospheric conditions and a discussion is needed here.

P19-22: Section 3.1.2: A summary of the actinic flux total uncertainties was expected here. Uncertainties were discussed for the calibrations, but what are the measured uncertainties? What is the impact of the improved UV stray light determination that comprises much of the analysis in this paper and how does it improve over previous

evaluations (Jakel et al., etc) . Also, what is the impact of stray light near detection limits where the uncertainty due to stray light determination is much larger (as a function of SZA, ozone, atmospheric conditions).

P22, L17-18: I recognize the complexity of the optical uncertainties (or biases), but I do think a greater summary of the Lohse and Bohn, 2017 results would be appropriate here. They are critical to evaluation of the data and understanding the total measurement uncertainties. This would not hinder the more detailed and quantitative explanations I expect are in the separate publication.

Technical comments:

P1 L13-14: Stray light is not strictly noise but rather a bias that varies with solar intensity. Reword.

P1 L7: "1×10ˆ10 cm−2s−1nm−1", Units should be photons cm-2 s-1 nm

P10, L5-6: I believe this should read "the SNR increases with the square root of the integration time and also improves at the shorter lamp distance"

P10, L9: Expand PTB to Physikalisch-Technische Bundesanstalt

P11, Fig 5 caption: Change to ". . .stray light signals were also subtracted"

P12, L29: Remove "also"

P13, L11: Remove "also"

P22, L2: Remove "Evidently, also"

---

## Author Comment (AC1) · 4 Jul 2017

**Reply to comments by Referee #1**

*This manuscript describes the calibration procedures and data evaluation of a CCD spectroradiometer (CCD-SR). The instrumental characteristics are partly compared with results from other publications. The performance of the CCD-SR is demonstrated for airborne and ground-based observations. The manuscript is very technical and gives a lot of details. It is well-written but could be shortened at some points, since it is no new instrumental techniques described here. The benefits for the reader should be pointed out more clearly in the introduction. What is new compared to former publications? The manuscript is recommended for publication in AMT. Nevertheless, the following comments should be addressed first.*

Reply: We thank the referee for the positive and very detailed evaluation of the text. All comments (italic font) will be addressed in the following.

We agree that the text is quite long and contains a lot of technical details. We'll will try to shorten the text where recommended but also note that at many points the referee is asking for additional information and additional figures. So it is difficult reduce the length of the paper. Some additional figures and tables will be placed in a supplement. We believe that although the technique is not new, many details matter here to obtain optimum results and we would like to encourage other researchers to look into the details of their instrument properties as well.

In order to point out the benefits for the reader more clearly, the final part of the introduction will be extended following page 3, line 31:

"In particular, for laboratory calibrations we adopt a new correction factor for subtracted stray light signals obtained with cutoff filters and an optimization of spectral sensitivities using extended integration times. For the evaluation of field data we introduce atmospheric cutoff wavelengths from radiative transfer calculations to define safe, condition dependent wavelength ranges for stray light determination. In addition the precision of j(O1D) measurements was improved by excluding spectral actinic flux densities below the cutoff wavelengths. These procedures are thought as recommendations for other users of similar instruments in order to raise the awareness for important instrument properties and characterizations, to illustrate essential evaluation steps and to clarify current limitations. Careful attention is thought to improve data quality and reproducibility. To evaluate the approach, example data from a flight on HALO, as well as ground based comparisons with a double-monochromator reference instrument will be shown."

*General comments:*
*1. The CCD-SR shows good performance also for ground-based observations. The disadvantage of less sensitivity and impact of stray light in the UV spectral range can be avoided by careful calibrations and some corrections as written in the paper. The comparison with the DM-SR has shown sufficient agreement. This precise scanning instrument, however, has much lower time resolution than the CCD-SR which is not sufficient for fast changing atmospheric conditions (as cloudy sky). For this reason I suggest to change the title: "Calibration and evaluation of CCD spectroradiometers for ground-based and airborne measurements of spectral actinic flux densities". There is no need to restrict the application*

*of the CCD-SR to airborne measurements. That could be made clear by extending the ground-based comparison (see comment #41).*

Reply: The focus on airborne operations was based on the specific requirements that can probably only be met by the CCD-SR. Of course the instruments are also useful for measurements on the ground and the procedures are similar. We'll change the title as recommended.

*2. Since the angular response of the entrance optics is highly affecting the quality of the data it would be worthwhile to implement it in this paper. In this case some changes to the text and text cuts should be made not to overload the manuscript when adding the discussion about the angular response. That would give the reader a full description of the instrumental performance in one paper.*

Reply: The second paper will be quite long by itself and the current paper would clearly be overloaded. Substantial text cuts are difficult as mentioned above. We did that separation not only to keep the individual papers short but also because we are dealing with two separate aspects of the measurements. The optical receiver characterizations and corrections are independent of the type of spectroradiometer used. On the other hand, most aspects of the current paper are specific for the CCD spectroradiometers and independent of the receivers.

We'll rephrase the sentence in the abstract on page 1, line 18 to clarify this: "Because optical receiver aspects are not specific for the CCD spectroradiometers they were widely excluded in this work and will be treated in a separate paper in particular with regard to airborne applications."

And we'll add the following sentence on page 3, line 6 to explain this: "Since these corrections are complex and independent of the type of spectroradiometer, we attend to this difficulty...."

We understand that for an assessment of total uncertainties more information on receiver specific correction factors is important. As a compromise, and as suggested by referee #2, we will show corrected and uncorrected spectra in the revised version of Fig. 11 (aircraft data) and more explicitly state the (low) extent of the corrections for the ground based measurements.

*Specific comments:*

*3. Give wavelength range of the CCD-SR in the abstract.*

Reply: The range will be specified (280-650 nm).

*4. p1l7: "flux densities in a 300 nm range" – What does it mean? Is it the detection limit at 300 nm wavelength?*

Reply: Yes, we'll change to " ... flux densities at wavelengths around 300 nm"

*5. p1l17: "below cutoff wavelength" – Not knowing the manuscript, the reader will not know what "cutoff wavelength" means. Better write something like that "below atmospheric cutoff wavelength which was simulated depending on solar zenith angle, ozone column, : : :"*

Reply: That was explained already in lines 8-9. We'll add the term "atmospheric" to clarify.

*6. p1l17: "with reference instrument data" – Which instrument? Tell the reader that is a high sensitive double monochromator.*

Reply: On page 1, line 11, where this instrument is mentioned for the first time we'll change to: "... comparisons with a double-monochromator based, highly sensitive reference spectroradiometer." On page 1, line 17 we'll specify: "...from linear regressions with data from the double-monochromator reference instrument."

*7. p1l19: "Overall, the investigated instruments are clearly : : :" – already mentioned in the beginning of the abstract (l3).*

Reply: The sentence will be removed.

*8. p2: Please discuss the dependence on temperature and pressure of molecular parameters.*

Reply: We'll extend the paragraph accordingly on page 2, line 27: "A further advantage of spectroradiometry is that the temperature and pressure dependencies of photolysis frequencies are obtained directly by taking into account the respective dependencies of the molecular parameters. This is particularly important for aircraft measurements where ambient conditions are most variable."

*9. p3l1: Explain shortly what is meant by "stray light". Furthermore, mention the lower sensitivity in the UV-B.*

Reply: We'll change the sentence accordingly: "(ii) the accuracy of measurements in the UV-B range that can be affected by low detector sensitivities and non-regularly reflected radiation within monochromators (stray light)."

*10. p3l3: "upward radiation in the UV range can often be neglected" – only valid over dark surfaces*

Reply: We agree that the statement was too sloppy as was also noted by referee #2. We'll use the following statement instead: "In contrast to ground-based operations where measurements of upward radiation in the UV range may be dispensable under conditions of low ground albedos, aircraft deployments require separate measurements in the upper and the lower hemisphere."

*11. p3l14: "have low time resolution" – What means low? Please give numbers.*

Reply: We will specify a range "...on the order of 0.5-2 min".

*12. p5l6: "The idea is to piece together : : :" – I am wondering if there is mismatch between the spectra sampled over different integration times, because the measurement volume is different. In particular for fast changing atmospheric conditions in combination with the speed of HALO, I would expect some differences. Did the authors compare the spectra for some examples without oversaturation? Furthermore, the linearity of the detectors should be shortly discussed.*

Reply: No systematic comparison was made of all spectra recorded quasi-simultaneously with different integration times. The successive measurements together take about 0.5 s. A significant change within this period of time could indeed produce differences. We checked the more variable upward flux densities of the example flight of 19 Dec (Fig. 14) and selected the longest possible wavelength where no saturation occurred at all integration times (321 nm). Significant differences between 100 ms and 300 ms data occurred only at a single point in time (around 12:01) caused by a flight maneuver where roll angles exceeded 10°. A similar check with a similar result was made for a (rare) flight where HALO was apparently flying within clouds, with high and strongly variable upward and downward flux densities of similar levels. We conclude that except for extremely rare circumstances differences between successive measurements with different integration times are insignificant. Regarding the linearity of the detector we refer to another section (see also point 24.). The following sentence will be included:

"This optimization is useful as long as integration times are short compared to the time scale of changes of measured flux densities. Moreover, the linearity of the CCD detector is a further requirement that can be tested in the laboratory (Sect. 2.2.3)."

*13. p5l8: ".. calculation of final actinic flux : : :" – In fact, these data are only a kind of quicklook. Post-processing is required as mentioned in the subsequent sentence.*

Reply: We'll skip the word "final" to avoid the impression that post-processing is optional. Moreover, in the succeeding sentence we switch from "many" to "some" applications.

*14. p5l12: Maybe a flow chart of the following calibrations would be helpful.*

Reply: Producing a flow chart for the calibration procedure turned out to be extremely difficult because it either oversimplified the procedure or even complicated it if every detail were included. For example an activity like "subtract background" does not explain that this background should be an average over 100 single measurements and that it should be made before and after lamp measurements and checked that there were no changes and that it should be done for all integration times etc. So we refrained from adding a flow chart.

*15. p5l22: "dark spectra" – Explain shortly.*

Reply: We extended the sentence: ".... dark spectra obtained upon covering optical receivers, ..."

*16. p5l23: "were fitted with an " – Why did the authors did not use also the Neon emission lamp which covers the longer wavelengths greater than 546 nm?*

Reply: We made (less frequent) checks with a HeNe laser at 632 nm. The hint with the Ne lamp is very useful. We'll purchase such a lamp for routine measurements in the future. The results of the HeNe laser measurements will be included in Tab. 1 and Fig. 2.

*17. p9l4: "Noise levels were found to increase with the square root of signals : : :" – Can you show a plot for illustration, maybe for different wavelengths?*

Reply: We produced a corresponding plot for dark and lamp measurements but do not distinguish between different wavelengths because all CCD pixels behave similarly. Noise is determined by the signal and signal only indirectly by wavelength through the spectral lamp output and the instrument sensitivity. Because we are dealing with (at least) three types of noise that add up geometrically, things are a bit complicated. For clarity, additional symbols e.g., for the noise at zero integration time ($N_r$), thermally and radiation induced shot noise ($N_d$, $N_s$) and for the electronic background signal ($S_{dark,0}$), will be introduced in the text and additional formulas will be inserted to explain the noise contributions. The plot is shown in FigD1.1. By the way: In the caption of Fig. 3 there was an error. The statement: "... and a dark current that increases with the square root of the integration time..." should read "...and a dark current that increases linearly with integration time."

*18. p10l1: "example of signal-to-noise ratios" – Give equation. I am wondering if it is correct to illustrate the wavelength dependence of the SNR this way. Since the SNR depends on the magnitude of the signal itself, a constant signal should be used for all wavelengths (however feasible) to demonstrate the pure wavelength dependence. As suggested in comment #17, plot the SNR as a function of the signal for different wavelengths.*

Reply: We'll include an equation how the SNR shown in the figure were calculated. Maybe the figure was not explained properly. It merely shows the SNR obtained during a calibration and the improvement obtained for close measurements and long integration times. We think this is explained sufficiently in the text. We'll include additional information in the figure caption:

"Signal-to-noise ratios (SNR) of 1000 W irradiance standard measurements with instrument 62001 as a function of wavelength for different integration times according to Eq. 7. (a) lamp distance 700 mm, (b) lamp distance 350 mm. The SNR is determined by the spectral lamp output and the instrument's spectral sensitivity that together produce the signal height and the corresponding noise (Fig. 4). The improvements for the close measurements are most useful for wavelengths below 400 nm."

*19. p10l8: "Laboratory calibration" – include radiometric*

Reply: Will be included.

*20. p10l16 "shows little wavelength dependence" – What means little, give numbers.*

Reply: We'll specify the maximum dependence as "(<2 mm, 300-650 nm)".

[Figure]

**FigD1.1:** Increase of noise ($N$) with the square root of CCD signals from dark measurements ($S_{dark}$) and 1000 W irradiance standard measurements ($S_{lamp}$) of instrument 62001. (a) dark, (b) lamp distance 700 mm (far), and (c) lamp distance 350 mm (close). Noise was corrected for residual noise ($N_r \approx 7$) at zero integration time and signals were corrected for constant electronic offsets ($S_{dark,0} \approx 900$). For the dark measurements at integration times ≤30 ms the noise increase is too small to be determined correctly. The dashed lines indicate an approximate linear dependence

*21. p11 Fig. 5: What is the gain of information showing the procedure for two integration times? The reader might be more interested in the stray light fraction than showing each step of the data evaluation.*

Reply: By showing two integration times in Fig. 5 we intended to demonstrate that UV-B radiation and stray light signals can be determined most accurately with the longest integration time and that for longer wavelengths shorter integration times are necessary

because of saturation. We agree that (though already visible in the second panels) the fraction of stray light can be plotted separately as an indicator for the stray light rejection of an instrument. We'll remove the left hand panels of Fig. 5, mention the saturation in the caption, and plot the stray light fraction in the fourth panel instead. The text and figure caption will be revised accordingly, see FigD1.2.

[Figure]

**FigD1.2:** Example signals obtained during laboratory calibrations of instrument 62001 with two cutoff filters at $\Delta t$=1000 ms and close lamp distance. The wavelength range relevant for the determination of stray light signals is zoomed in. Saturation occurred around 400 nm for this integration time. (a) Total signals of lamp radiation with no filter ($S_{lamp}$), of lamp radiation with WG320 filter ($S_{filter\ 1}$), of lamp radiation with WG360 filter ($S_{filter\ 2}$), and dark signals ($S_{dark}$). (b) After subtraction of dark signals, stray light signals were estimated by linear regressions in a range 270-300 nm and extrapolated over the whole spectral range (dashed lines) before final subtraction. (c) Signal contribution of stray light in the atmospherically most relevant wavelength range. Similar figures for the other instruments are shown in the Supplement.

*22. p12l12: "WG320, WG360" – Please refer to Schott, otherwise it is not clear where the "WG" comes from.*

Reply: Schott was mentioned page 10, line 21. We'll change to (Schott, White Glass) to explain the "WG".

*23. p12l23: ": : : stray light quickly diminishes with increasing wavelength: : :" As mentioned in comment #21, this would be good to see in a plot for all spectroradiometers.*

Reply: We'll provide similar figures as the revised version of Fig. 5 (FigD1.2) for the remaining four instruments in supplementary information.

*24. p13 Fig. 6: Here, a test of the linearity could be implemented.*

Reply: We will include a figure (see FigD1.3) that demonstrates the linearity. On page 14, line 2, the following paragraph will be included:

"The relation in Eq. 10 was tested by comparing measured sensitivities for an integration time of 10 ms and calculated sensitivities for the same integration time from measurements at 30, 100, 300 and 1000 ms as shown in Fig. D1.3. Sensitivities at 10 ms were chosen as an example because no saturation occurred, i.e. the comparison is covering all wavelengths (but not at all integration times). Linear regressions produced slopes deviating less than 1% from unity confirming the strict linearity of the measurements. Equation 10 was therefore used to derive consistent, optimum sensitivities for all integration times, ..."

*25. p13l14: It is suggested to delete the paragraph. After introducing the spectral calibration, the reader assumes that the wavelength shift is considered in the radiometric calibration.*

Reply: The paragraph will be deleted.

*26. p14l11: Also here, delete the paragraph. The reader will not be interested in the Metcon software.*

Reply: Not sure this applies for all readers but we agree that this information is not essential. The paragraph will be deleted.

*27. p16l14: "in total uncertainties between : : :" - The error due to angular response is not included here as it should be when talking about the total uncertainty.*

Reply: In this section we are talking about the total uncertainties of the spectral calibrations. We'll add the term "spectral" to the title of the section, i.e. "spectral calibration uncertainties", and in the final sentence of the section. Moreover we'll add the following paragraph for clarification and will address the total uncertainties in the final sections as also recommended by referee #2:

"These uncertainty estimates were derived from carefully controlled laboratory measurements at normal incidence of radiation and roughly apply for any CCD spectroradiometer with similar properties. However, total atmospheric measurement uncertainties can be affected by additional factors related with receiver specific angular response imperfections, atmospheric stray light influence and instrument noise that are

dependent on measurement conditions. Based on the laboratory characterizations, the influence of instrument noise on detection limits and measurement precisions is estimated in the following before atmospheric measurements are addressed in Sect. 3."

[Figure]

**FigD1.3:** Comparison of spectral sensitivities of instrument 62001 for an integration time of 10 ms ($\Delta t_2$) with scaled spectral sensitivities obtained at other integration times ($\Delta t_1$). Scaling factors were calculated according to Eq. 11. Data points cover all wavelengths. The dashed line indicates perfect linearity.

*28. p17l12: "Detection limits are usually defined as the three-fold : : :" – Please give a reference.*

Reply: We will include a reference and replace "defined" by "estimated" to indicate that this is not a strict rule for all types of measurement.

*29. p17l20: "the noise limits correspond to 0.1% and 0.0001%..." – Are these values derived from the table? Give the reference (typical maximum) values. Tab. 3 lists also noise equivalent photolysis frequency for a limited spectral range above 300 nm in brackets. A better motivation might be to give the results for two situations (low SZA, low total ozone column vs. high SZA, high total ozone column). Note the conditions in the table caption. It would give the reader a smooth introduction to the cutoff wavelength paragraph.*

Reply: (i) The statement (0.1% and 0.0001%) was just a rough estimate to rate the detection limits in terms of maximum atmospheric values. We'll give the corresponding maximum values in the text as $4\times10^{-5}s^{-1}$ and $1\times10^{-2}s^{-1}$. By the way, there was a typo in Tab. 3. For the 1000 ms integration time the exponent of the first value should read −8 instead of −9 for $j(NO_2)$.

(ii) The detection limits were derived for conditions with no radiation. Obviously, the use of cutoff wavelengths mostly pay off under conditions with low radiation as is also evident from Fig. 13. What the referee is asking for is the noise-induced precision of the $j$ measurements under various atmospheric conditions and whether or not the use of the cutoff wavelengths leads to improvements. We estimated this by using the simulated spectra and the increase of noise with instrument signals which is now explained in more detail in the preceding sections (see point 17.). The following new section will be included:

**Measurement precisions**
"The influence of radiation induced shot noise on the precision of $F_\lambda$ measurements as well as the effect of cutoff wavelengths on the precision of $j(O^1D)$ and $j(NO_2)$ under various atmospheric conditions were investigated based on the same simulated clear-sky downward $F_\lambda$ spectra that were used to derive the cutoff wavelengths. Signal spectra for different integration times were calculated from the $F_\lambda$ by multiplication with the spectral sensitivities of instrument 62001 (Fig. 6). The corresponding noise was obtained according to {FigD1.1} and optimized noise spectra were combined by preferring long integration times unless saturation levels were reached. A maximum 300 ms integration time was assumed as during atmospheric measurements (Sect. 3.1.2) and the respective noise equivalent $F_\lambda^{NE}$ were derived. The results are listed in TabD1.1 together with the $F_\lambda$ from the model for a number of solar zenith angles and wavelengths at an altitude of 15 km. With increasing signals, shot noise increases and shorter integration times become necessary. Accordingly, the $F_\lambda^{NE}$ increase with wavelength and solar elevation. A comparison of $F_\lambda$ and $F_\lambda^{NE}$ shows that except for the shortest wavelength, high signal-to noise ratios can be expected under all conditions ($\geq$600). For a zero-spectrum the results obtained with the measured dark spectra in Fig. 9 for an integration time of 300 ms were reproduced. The corresponding data for an altitude of 0 km can be found in the Supplement. The potential influence of stray light signals is not considered in this analysis. However, as will be shown in Sect. 3, stray light induced shot noise is very limited. By repeatedly applying random $F_\lambda^{NE}$ noise for each pixel, simulated precisions of photolysis frequencies for instrument 62001 were obtained. These data are listed in TabD1.2 for the same conditions as in TabD1.1. For $j(O^1D)$ the precision is almost constant and independent of the photolysis frequency because radiation induced shot noise is apparently secondary. The application of cutoff wavelengths from the lookup tables led to significant improvements in particular towards large SZA because of increasing cutoff wavelengths. In contrast, for $j(NO_2)$, photon induced shot noise plays an important role and several shorter integration times were involved in the simulated measurements. Accordingly, the absolute noise increases with increasing $j(NO_2)$. The application of cutoff wavelengths led to no changes for $j(NO_2)$, the results were therefore not included in Tab. D1.2. For a zero-spectrum the results obtained with the measured dark spectra for an integration time of 300 ms were again reproduced (Tab. 3). Corresponding data for an altitude of 0 km can be found in the Supplement."

*30. p17l24-p17l31: This paragraph could be deleted. It was already stated in the introduction that in the UV-B the ozone photolysis is more affected than the NO2-photolysis.*

Reply: The statement in the introduction is less specific. Here we explain why instrument noise below 300 nm affects the detection limits of $j(O^1D)$ while $j(NO_2)$ is unaffected. This is important to understand the influence of cutoff wavelengths on $j(O^1D)$."

**TabD1.1:** Downward spectral actinic flux densities $F_\lambda$ from radiative transfer calculations for selected wavelengths and solar zenith angles for an altitude of 15 km and an ozone column of 300 DU (left) and simulated noise equivalent actinic flux densities $F_\lambda^{\mathrm{NE}}$ of instrument 62001 for a maximum 300 ms integration time (right). The entry SZA$>$100° indicates dark conditions.

| $\lambda$ / nm
 SZA / deg | 300 | 350 | 400 | 450 | 500 | 550 | 600 | 650 | 300 | 350 | 400 | 450 | 500 | 550 | 600 | 650 |
|---|---|---|---|---|---|---|---|---|---|---|---|---|---|---|---|---|
| | $F_\lambda$ / $10^{12}\mathrm{cm}^{-2}\mathrm{s}^{-1}\mathrm{nm}^{-1}$ | | | | | | | | $F_\lambda^{\mathrm{NE}}$ / $10^{10}\mathrm{cm}^{-2}\mathrm{s}^{-1}\mathrm{nm}^{-1}$ | | | | | | | |
| 0 | 5.8 | 200 | 360 | 500 | 490 | 520 | 530 | 510 | 2.2 | 29 | 52 | 53 | 50 | 54 | 58 | 61 |
| 30 | 4.0 | 200 | 360 | 500 | 490 | 510 | 520 | 510 | 1.9 | 28 | 52 | 53 | 50 | 54 | 57 | 61 |
| 50 | 1.5 | 200 | 360 | 490 | 480 | 510 | 520 | 510 | 1.4 | 28 | 52 | 52 | 50 | 53 | 57 | 60 |
| 60 | 0.50 | 190 | 350 | 490 | 480 | 500 | 510 | 500 | 1.1 | 28 | 52 | 52 | 50 | 53 | 57 | 60 |
| 70 | 0.06 | 180 | 340 | 470 | 470 | 490 | 490 | 490 | 1.0 | 27 | 51 | 51 | 49 | 52 | 55 | 59 |
| 80 | $0.55^{a}$ | 140 | 290 | 430 | 430 | 440 | 430 | 460 | 1.0 | 24 | 47 | 49 | 47 | 50 | 52 | 57 |
| 84 | $0.32^{a}$ | 100 | 240 | 380 | 380 | 380 | 360 | 410 | 1.0 | 11 | 24 | 46 | 45 | 46 | 48 | 55 |
| 88 | $0.14^{a}$ | 26 | 93 | 200 | 210 | 190 | 160 | 250 | 1.0 | 3.2 | 15 | 19 | 34 | 19 | 18 | 25 |
| $>$100 | 0 | 0 | 0 | 0 | 0 | 0 | 0 | 0 | 1.0 | 0.6 | 0.4 | 0.3 | 0.3 | 0.3 | 0.3 | 0.4 |

$^{a}$ $F_\lambda$ / $10^{10}\mathrm{cm}^{-2}\mathrm{s}^{-1}\mathrm{nm}^{-1}$

**TabD1.2:** Photolysis frequencies from radiative transfer calculations of downward spectral actinic flux densities for selected solar zenith angles at an altitude of 15 km and an ozone column of 300 DU (left) and simulated noise equivalent photolysis frequencies of instrument 62001 for a maximum 300 ms integration time (right). $j(\mathrm{O}^1\mathrm{D})$ precisions in brackets were obtained by applying variable cutoff wavelengths (FigD1.4). The entry SZA$>$100° indicates dark conditions with zero spectral actinic flux densities.

| SZA / deg | photolysis frequency | | noise equivalent photolysis frequency | |
|---|---|---|---|---|
| | $j(\mathrm{O}^1\mathrm{D})$ / $\mathrm{s}^{-1}$ | $j(\mathrm{NO}_2)$ / $\mathrm{s}^{-1}$ | $j(\mathrm{O}^1\mathrm{D})$ / $\mathrm{s}^{-1}$ | $j(\mathrm{NO}_2)$ / $\mathrm{s}^{-1}$ |
| 0 | $6.09\times10^{-5}$ | $9.56\times10^{-3}$ | $1.1\times10^{-7}$ ($3.6\times10^{-8}$) | $1.2\times10^{-6}$ |
| 30 | $4.99\times10^{-5}$ | $9.50\times10^{-3}$ | $1.0\times10^{-7}$ ($3.0\times10^{-8}$) | $1.2\times10^{-6}$ |
| 50 | $3.20\times10^{-5}$ | $9.27\times10^{-3}$ | $1.0\times10^{-7}$ ($2.1\times10^{-8}$) | $1.2\times10^{-6}$ |
| 60 | $2.13\times10^{-5}$ | $8.98\times10^{-3}$ | $1.0\times10^{-7}$ ($1.5\times10^{-8}$) | $1.2\times10^{-6}$ |
| 70 | $1.09\times10^{-5}$ | $8.38\times10^{-3}$ | $1.1\times10^{-7}$ ($9.7\times10^{-9}$) | $1.2\times10^{-6}$ |
| 80 | $3.01\times10^{-6}$ | $6.81\times10^{-3}$ | $1.1\times10^{-7}$ ($5.7\times10^{-9}$) | $9.7\times10^{-7}$ |
| 84 | $1.14\times10^{-6}$ | $5.25\times10^{-3}$ | $1.1\times10^{-7}$ ($4.7\times10^{-9}$) | $7.2\times10^{-7}$ |
| 88 | $1.42\times10^{-7}$ | $1.68\times10^{-3}$ | $1.0\times10^{-7}$ ($3.7\times10^{-9}$) | $2.7\times10^{-7}$ |
| $>$100 | 0.0 | 0.0 | $1.0\times10^{-7}$ ($3.6\times10^{-9}$) | $2.4\times10^{-8}$ |

*31. p17l32: "cutoff-wavelength" - atmospheric cutoff wavelength. The authors created some look-up-tables (as mentioned on p18l3). A contour plot showing the dependence of the cutoff wavelength on SZA and total ozone column would be nice to see.*

Reply: A contour plot will be included for an altitude of 15 km (FigD1.4), a corresponding plot for 0 km will be shown in the Supplement.

*32. p19l14: How are temperature and pressure variations considered in the look-up tables? There should be some sensitivity in particular for ozone photolysis.*

Reply: Temperature and pressure variations were not considered. The cutoff wavelengths were defined by a limiting flux density (page 17, line 33). In the range of the cutoff wavelengths O($^1$D) quantum yields are independent of temperature and cross sections merely go down slightly at lower temperatures. In any case it is ensured that the fraction of $j(\mathrm{O}^1\mathrm{D})$ that can be attributed to wavelengths below the cutoff is insignificant (page 18, line 6).

*33. p19l21-p21l23: This section can be shortened. A flow chart summarizing the steps to create a spectrum from raw data might be helpful.*

Reply: We revised the figures which shortened the page space of the section, see 35.-37. For a flow chart the same problem arises as for the calibration procedure. It is either too simple if no additional information is given for each action, or too complicated.

[Figure]

**FigD1.4:** Contour plot of atmospheric cutoff wavelengths (nm) for an altitude of 15 km as a function of solar zenith angles (SZA) and ozone columns. The data were derived from radiative transfer calculations of downward clear sky spectral actinic flux densities defining a lower limit $F_\lambda \leq 5 \times 10^9$ cm$^{-2}$ s$^{-1}$ nm$^{-1}$. A similar plot for an altitude of 0 km can be found in the Supplement.

*34. p19l25 Is the dark signal taken from laboratory measurements? If yes, how are changes be considered during the campaign (p8l1)?*

Reply: We used the dark measurements from the lab measurements because conditions were more stable. Comparisons of dark signals before and after deployments so far showed no significant differences, i.e. standard deviations of ratios were within 0.1-0.3% indicating that the spectral structure is very stable. Absolute variations are corrected for together with the stray light signal as described on page 8, line 1-8 and page 19, line 27-28. We'll add the information that dark signals from the lab were taken.

*35. p20 Fig. 10: The upper two panels are sufficient to illustrate the stray light and offset correction for field measurements. An additional subfigure showing the stray light fraction would be more interesting than the remaining six panels (supporting also p22l2).*
*36. p21 Fig. 11: Combine with Fig.10., the logarithmic representation would be sufficient. Could the authors add a comparison with simulations for the downward actinic flux density?*
*37. p22l8: "starting with the longest available: : :" – As mentioned earlier the reader might be more convinced of this combination method by showing spectra measured for different*

*integration times in one plot with indication at which wavelengths the final spectrum were merged.*

Reply to points 35-37: We revised Fig. 10 and 11 but kept them separated. In Fig. 10 the two uppermost panels remain unchanged, the additional three panels for each instrument were removed and a second panel was included showing the contributions of stray light, as requested (FigD1.5). In Fig. 11 we now also show two panels each where linear and semi-logarithmic representations were combined. The linear plots more clearly show the differences between the upward and the downward flux densities. The upper panels demonstrate how final spectra were put together from different integration times. The lower panels show a comparison of uncorrected, corrected (influence of optics) and simulated spectra, as requested (FigD1.6). The corresponding text and figure captions will be revised accordingly.

[Figure]

**FigD1.5:** Examples of flight raw data and evaluations of instruments 62000 (lower hemisphere, left) and 62001 (upper hemisphere, right). Data were obtained during a HALO flight on 20 Dec 2013 17:30 UTC over the North Atlantic (15.0N, 55.9W, 13.2 km) under conditions with few scattered low-lying clouds (solar zenith angle 47°, ozone column 245 DU). In panels (a) and (b) different colors indicate the evaluation steps: raw data (red), background subtraction (green) and stray light subtraction (blue). Stray light signals (dashed black lines) were determined by linear regression of background corrected signals in a range 270 nm to 291 nm (cutoff wavelength). In panels (c) and (d) the contributions of the inter- and extrapolated stray light signals are shown for the integration times eventually used in the displayed, most effected and atmospherically relevant wavelength range >280 nm.

[Figure]

**FigD1.6:** Evaluated upward (left) and downward (right) spectral actinic flux densities of the data shown in FigD1.5 in linear and semilogarithmic representations. Arrows point to the respective axes. In panels (a) and (b) spectra obtained with different integration times are plotted upon each other. Because already for 30 ms (a) and 10 ms (b) no saturation occurred, the data shown for the shortest integration time in each panel were not used for the final optimization of spectra. In panels (c) and (d) the spectra denoted as uncorrected (red) represent the optimized spectra of panels (a) and (b). Optical receiver specific corrections led to the slightly modified, corrected spectra (blue). Results of radiative transfer calculations for clear-sky conditions are denoted as modeled (green).

*38. p23l3: The reader probably expects the results of the photolysis frequency right here.*

Reply: Sections 3.2 and 3.3 will be exchanged. In the abstract the final sentence will be deleted and "on HALO" included on page 1, line 10 instead.

*39. p23l16: One example is sufficient, either 62000 or 62001. When comparing with DM-RS add the measurement uncertainty for both instruments in Fig. 12.*

Reply: In the revised version we'll only show data of instrument 62001 in a single-column width figure. Error bars will be included. See FigD1.7.

*40. p24 Fig.12: The lower left panel would be sufficient. Could the authors here also some simulated spectrum?*

Reply: We prefer to show both linear and semi-logarithmic plots. The linear plots more clearly show any difference at greater values and only in the linear representation error bars from the calibration are visible. The semi-log plot emphasizes the low values very strongly

and only here the noise induced uncertainties become apparent. A simulated spectrum will be included in all plots (FigD1.7).

[Figure]

**FigD1.7:** Comparison of actinic flux density spectra obtained on the ground with a double-monochromator based reference instrument (DM-SR) (red) and instrument 62001 (blue). Measurements were made on 01 Aug 2013 at Jülich (Germany) under clear-sky conditions. The spectra were taken around 12:00 UTC at a solar zenith angle of 33° and an ozone column of 310 DU. Black data points show the results of radiative transfer calculations for the same conditions. The different representations emphasize the increase of actinic flux densities in the UV-B range in panel (a) and the dynamic range of data in panel (b). Green data points in the semi-logarithmic plots of panel (c) show negative values that were plotted at their absolute values to make them visible. Full lines with minimum symbol size indicate the corresponding upper or lower limits after addition or subtraction of instrument noise. The dashed vertical line shows the cutoff wavelength below which values of the CCD-SR were normally set to zero. Note the different spectral actinic flux density and wavelength ranges.

*41. p25l1: The authors compare j-values derived from CCD-SR and DM-SR for two days including cloud periods. Here the advantage of the better time resolution could be directly shown by a time series for of j-values with best temporal resolution for each instrument. Furthermore, filtering these cloud events from Fig. 13 might be worthwhile to exclude the data sampled during variable conditions within the time frame of one DM-SR measurement.*

Reply: In the supplement we will show the time series of data from two days, one with broken clouds and one clear-sky day. For both days we will also show selected 15 min periods where the effects and no-effects of the different time resolutions will become visible. In addition a plot corresponding to the right hand side of Fig. 13 with data from the clear-sky day will be presented instead of the three days comparison in the main paper. However, in the main paper we prefer to stick to the current version for instrument 62001 because it corresponds to the routine procedure before or after a deployment. Clear-sky days are rare in Jülich and waiting for the perfect conditions is no option. Moreover, despite the timing-problems, the comparisons should cover clear and cloudy periods.

*42. p29 Fig. 14: Show a map instead of the time series of latitude and longitude. Combine SZA and total ozone column in one panel. Remove date in x-axis in the lower panels. Add some simulated j-values for downward components.*

Reply: The figure will be revised as recommended and a map will be included. (FigD1.8 and FigD1.9).

[Figure]

**FigD1.8:** Map of the flight route of HALO on 19 Dec 2013 from Oberpfaffenhofen, Germany to the island of Barbados. Indicated times are UTC.

*Technical comments:*

*43. p2l13, p2l22: "spectral actinic photon flux density" – delete "photon"*

Reply: Because actinic flux densities can be "photon" as well as "energy" flux densities and the formula is only valid for the first, we intended to clarify this by adding the term "photon" to the variable name. We will remove "photon" (also on page 2, line 22 and page 12, line 4) and add a sentence for clarification following page 2, line 17: "$F_\lambda$ is inserted in corresponding molecular units ($cm^{-2}s^{-1}nm^{-1}$)".

[Figure]

**FigD1.9:** Example data from a HALO research flight on 19 Dec 2013 from Oberpfaffenhofen (Germany) to Barbados. Altitude (ALT) and static temperature (TMP) in panel (a), as well as solar zenith angles (SZA) and total ozone columns (TOC) in panel (b), are important boundary conditions and parameters used in the data evaluation. Photolysis frequencies $j(O^1D)$ and $j(NO_2)$ in panels (c) and (d) were calculated for the corresponding static temperatures. Rapid fluctuations of upward components of photolysis frequencies were induced by underlying clouds. Modeled data are clear-sky upward and downward components of photolysis frequencies from radiative transfer simulations.

*46. p3l16: "maneuvers in flight" - flight maneuvers*

*47. p4l7: "properties" – characteristics*

*48. p4l14: "lengths" – length*

*49. p5l6: "The idea is to piece together : : :" – combine*

*50. p5l12: "characterisation" – characterization*

Reply: Points 44-50 will be corrected as recommended.

*51. p6 Fig. 2: Rearrange plot to save some space. Maybe one combined color coded plot is sufficient showing the spectra as function of wavelength difference referred to center wavelength (+/- 4 nm). Otherwise, please give labels for each subfigure (a) –(e).*

Reply: We'll go for the second option. Even with different colors, all lines in a single plot are hard to distinguish.

*52. p8 Fig. 3: Unit of signal? Counts?*

Reply: We avoid using "counts" as a unit. The recorded signals are dimensionless "data numbers" or "analog-to-digital units". If "*S*/counts" were used, also the unit of spectral

sensitivities would become "counts $cm^2$ s nm" (or "counts/photons $cm^2$ s nm" if we would use another qualifying term). To keeps things simple and to avoid confusion with "photon counting" we will explain the nature of the signals in more detail, following page 5, line 1: "CCD data acquisition is controllable by purpose-built software provided by Metcon. The recorded signals (*S*) are dimensionless 16-bit signal counts or so-called analog-to-digital units (ADU) ranging between 0 and 65535."

*53. p8l13: "is considered a combination" - "is considered as a combination"*

Reply: Will be changed.

*54. p9 Tab. 2: Numbers in Counts?*
Reply: See point 52)

*55. p9 Fig. 4: Label subfigures and use larger points. Plot subfigures in two columns.*

Reply: We'll label the subfigures and increase the symbols. The two-row arrangement was chosen deliberately because it fits to the final column width of the journal (with two columns). Two-column figures are often downsized. Panels in all figures will be labelled.

*56. p12l8 and hereafter: Maybe better to use the index "stray" instead of "scat" in the S-variable.*

Reply: That was a leftover from a previous draft. The index "stray" will be used consistently throughout the text.

*57. p16 Fig. 8: Figure caption "hangar measurements" - field measurements, label the subfigures with (a) - (d)*

Reply: The figure will be modified accordingly, as well as the caption and the indices.

*58. p16l2: "Firstly by the : : :" – no sentence, link it to the sentence before by using ";"*

Reply: Will be changed as recommended.

*59. p17l14: "be considered a theoretical" - as a theoretical*

Reply: Will be changed.

*60. p17l18: "are listed in Tab. 2" - Tab. 3*

Reply: Corrected.

*61. p17l25: "Tab. 2" - Tab. 3*

Reply: Corrected.

---

## Author Comment (AC2) · 4 Jul 2017

**Reply to comments by Referee #2**

*Summary:*
*This paper characterizes the CCD-SR instrument that deploys on the HALO aircraft and continues the discussion of actinic flux measurements and calibrations using Metcon CCD-based spectrometers. The determination of spectral and wavelength assignment accuracies and sensitivities are discussed and a new technique for removing stray light was demonstrated to improve UV-B spectral accuracy and account for low limits of detection. Cutoff wavelengths are determined for each measured spectra based on radiative transfer modeled fluxes to reduce noise in photolysis products and examine stray light impacts. Ground-based comparisons with low-stray light instrumentation show strong agreement to near the detection limits. The resulting data are shown to be of sufficiently high accuracy for studies of actinic flux and calculated photolysis frequencies. This work is of value, well written and recommended for publishing after addressing the comments.*

Reply: We thank the referee for the positive evaluation of the text. All comments (italic font) will be addressed in the following.

*General comments:*
*P3 L3-4: The authors contend that "upward radiation in the UV range can often be neglected" at surface sites. Upwelling jNO2 is often 4-10% to the total photolysis and can be much greater. The upward UV radiation can rarely be neglected (though it can be estimated from the downwelling). This sentence should be dropped or adjusted.*

Reply: We agree that the statement was too sloppy as was also noted by referee #1. We will clarifiy this and use the following statement instead:

"In contrast to ground-based operations where measurements of upward radiation in the UV range may be dispensable under conditions of low ground albedos, aircraft deployments require separate measurements in the upper and the lower hemisphere."

*P12, L24-27: The authors may have some luck in that they found linear stray light structure in the UV-B in 4 of the 5 tested spectrometers (with the exception of 45853). The paper's entire stray light analysis relies on a linear-based subtraction. However, non-linear stray light response is not fully controlled for in the manufacturing and is determined by testing after production. Thus, I think they should state explicitly that determination of the stray light spectral structure is required to assess if a linear regression is appropriate.*

Reply: We agree that this should be stated more explicitly. Fig. 5 will be revised as shown in FigD2.1 below. Similar plots will be shown in the Supplement for all instruments. The corresponding paragraph was revised and extended following page 12, line 22:

[revised manuscript text omitted]

*P19, L13-15: The lookup tables are appropriate for clear-sky determinations of cutoff wavelengths, but no mention is made of the impacts of clouds and aerosols. Extinction by clouds and aerosols can greatly reduce the in situ measured flux and cutoff wavelength would be higher than expected from the table. In these cases, the variability in cutoff could be larger than 2 nm (as mentioned on P18 L5-6). Ideally the measured spectra could be evaluated for the cutoff, in concert with the model. The measurements are often most interesting in these complex atmospheric conditions and a discussion is needed here.*

Answer: The idea behind the cutoff wavelengths is that they should set a safe lower limit to the wavelength range where significant flux densities can be expected and a safe upper limit for the range in which stray light can be determined. By choosing simulations under clear sky conditions and selecting downward radiation we ensure that under all conditions "real" cutoff wavelengths will be equal or greater than those in the lookup tables. We do not want to exhaust the method by approaching most realistic cutoff wavelengths for each spectrum

but rather stay on the "safe side". On page 19, line 14 we'll include the following paragraph to clarify this:

"Even though the cutoff wavelengths were derived from simulated clear sky downward actinic flux densities, they will be applied in the following under all conditions as well as for upward actinic flux densities that are typically much lower. The data from the lookup tables were taken as safe lower limits for convenience. An unaccounted presence of clouds, for example, would shift cutoff wavelengths towards slightly greater values which is non-critical for the data analysis."

*P19-22: Section 3.1.2: A summary of the actinic flux total uncertainties was expected here. Uncertainties were discussed for the calibrations, but what are the measured uncertainties? What is the impact of the improved UV stray light determination that comprises much of the analysis in this paper and how does it improve over previous evaluations (Jakel et al., etc) . Also, what is the impact of stray light near detection limits where the uncertainty due to stray light determination is much larger (as a function of SZA, ozone, atmospheric conditions).*

Reply: We'll try to answer all these questions in a revised version of section 3.2.

(1) As a prerequisite, measurement precisions were simulated based on spectra from a radiative transfer model and the signal dependence of instrument noise obtained in laboratory measurements. Please refer to the answer of point 29 of referee #1 and tables TabD1.1, TabD1.2 shown there.

(2) In the previous version of section 3.2 on page 23, line 30, the following statement was made: "A closer look at the CCD-SR residual noise during day and night reveals that while in the dark the noise is consistent with the $F^{NE}$ obtained in the laboratory measurements (Fig 9), it is increased by a factor of up to 2–3 during daytime. This is attributed to additional noise induced by stray light." This statement was wrong because data were compared in a too wide wavelength range, still affected by atmospheric radiation.

(3) Section 3.2 was split up into three subsections. The first is dealing with actinic flux density spectra:

[revised manuscript text omitted]

(4) The influence of stray light signals on the uncertainties of photolysis frequencies will be addressed in a new paragraph in the new section on photolysis frequencies:

"In order to assess the estimated uncertainty of the stray light signal subtraction on photolysis frequencies, $j(O^1D)$ and $j(NO_2)$ were calculated from the subtracted stray light signals on a clear-sky day assuming again a 1% uncertainty for the inter- and extrapolated values. For $j(O^1D)$ the resulting total uncertainties strongly depend on whether or not data below cutoff wavelengths are taken into account. Without the help of cutoff wavelengths, total uncertainties are $0.06 \times j(O^1D) + 7 \times 10^{-7}$ s$^{-1}$ around noon and $1 \times 10^{-7}$ s$^{-1}$ after sunset (TabD1.2). With cutoff wavelength these numbers are $0.06 \times j(O^1D) + 1.3 \times 10^{-7}$ s$^{-1}$ around noon and $0.3 \times 10^{-9}$ s$^{-1}$ after sunset. Again it should be noted that despite greater absolute values at small SZA, the relative importance of these additional uncertainties increase with SZA reaching maxima of 3% and 100% around SZA=85° with and without cutoff wavelengths. Thus, the use of cutoff wavelengths expectedly also helps to confine the stray light influence. For $j(NO_2)$ the corresponding values are negligible under all conditions (<0.2%)."

(5) With regard to the methods of stray light determination the following paragraph was included following page 25, line 33:

"However, for instruments without UG5 filter this procedure is no option because the spectral shape of the stray light is strongly influenced by radiation blocked out by a 700 nm filter. Moreover, the slope of the stray light signals during lamp calibrations was usually slightly negative while in the atmosphere they were typically slightly positive. For that reason laboratory measurements were not consulted to estimate the shape of atmospheric stray light signals also because the color temperature of the calibration lamps is much lower than that of the sun. This may explain why the stray light contributions in FigD2.2 diminish much faster than those in FigD2.1 and corresponding figures in the Supplement. Fitting the spectral dependence of measured stray light signals in a condition dependent range defined by the atmospheric cutoff wavelengths was therefore a manifest approach. Because the results were satisfactory no alternative procedures were systematically tested. The practice is believed to be widely transferable to spectral irradiance measurements."

*P22, L17-18: I recognize the complexity of the optical uncertainties (or biases), but I do think a greater summary of the Lohse and Bohn, 2017 results would be appropriate here. They are critical to evaluation of the data and understanding the total measurement uncertainties. This would not hinder the more detailed and quantitative explanations I expect are in the separate publication.*

Reply: In the revised version of the paper we will bring out more clearly that we are dealing with two separate aspects of the measurements. We'll rephrase the sentence in the abstract on page 1, line 18 to clarify this: "Because optical receiver aspects are not specific for the CCD spectroradiometers they were widely excluded in this work and will be treated in a separate paper in particular with regard to airborne applications."

And we'll add the following sentence on page 3, line 6 to explain this: "Since these corrections are complex and independent of the type of spectroradiometer, we attend to this difficulty...."

We agree that for an assessment of total uncertainties more information on receiver specific correction factors is important. We will therefore show corrected and uncorrected spectra in

the revised version of Fig. 11 (aircraft data, see FigD2.4 below) and more explicitly state the (low) extent of the corrections for the presented ground based measurements. Because the discussion on uncertainties is already quite elaborate, more details on receiver related corrections under various conditions would be confusing.

[Figure]

**FigD2.4:** Evaluated upward (left) and downward (right) spectral actinic flux densities of the data shown in FigD1.5 in linear and semilogarithmic representations. Arrows point to the respective axes. In panels (a) and (b) spectra obtained with different integration times are plotted upon each other. Because already for 30 ms (a) and 10 ms (b) no saturation occurred, the data shown for the shortest integration time in each panel were not used for the final optimization of spectra. In panels (c) and (d) the spectra denoted as uncorrected (red) represent the optimized spectra of panels (a) and (b). Optical receiver specific corrections led to the slightly modified, corrected spectra (blue). Results of radiative transfer calculations for clear-sky conditions are denoted as modeled (green).

*Technical comments:*

*P1 L13-14: Stray light is not strictly noise but rather a bias that varies with solar intensity. Reword.*

Reply: We'll rephrase the sentence to "The spectra expectedly revealed increased daytime levels of stray light induced signals and noise below atmospheric cutoff wavelengths."

*P1 L7: "1×10ˆ10 cm−2s−1nm−1", Units should be photons cm-2 s-1 nm*

Answer: According to SI conventions qualifying terms like "photons" or "molecules" should not be part of a unit and in this case there is no risk it could be confused with any other unit. Referee #1 even asked us to remove "photon" from the name of the F-variable which we

sometimes use to clarify the nature of the flux densities we are talking about. But we agree that this is not essential. To make sure there is no confusion we inserted a clarification following page 2, line 17 where the quantity is introduced: "$F_\lambda$ is inserted in corresponding molecular units ($cm^{-2}s^{-1}nm^{-1}$)".

*P10, L5-6: I believe this should read "the SNR increases with the square root of the integration time and also improves at the shorter lamp distance"*

Answer: We agree that singular sounds better. We'll also change it in the lines above.

The following four points will be changed as recommended:

*P10, L9: Expand PTB to Physikalisch-Technische Bundesanstalt*

*P11, Fig 5 caption: Change to ". . .stray light signals were also subtracted"*

*P12, L29: Remove "also"*

*P13, L11: Remove "also"*

*P22, L2: Remove "Evidently, also"*